# Reprogramming of breast tumor–associated macrophages with modulation of arginine metabolism

Veani Fernando[1,2], Xunzhen Zheng[1], Vandana Sharma[1,3], Osama Sweef[4], Eun-Seok Choi[4], Saori Furuta[1,4]

**HER2+ breast tumors have abundant immune-suppressive cells, including M2-type tumor-associated macrophages (TAMs). Although TAMs consist of the immune-stimulatory M1 type and immune-suppressive M2 type, the M1/M2-TAM ratio is reduced in immune-suppressive tumors, contributing to their immunotherapy refractoriness. M1- versus M2-TAM formation depends on differential arginine metabolism, where M1-TAMs convert arginine to nitric oxide (NO) and M2-TAMs convert arginine to polyamines (PAs). We hypothesize that such distinct arginine metabolism in M1- versus M2-TAMs is attributed to different availability of $BH_4$ (NO synthase cofactor) and that its replenishment would reprogram M2-TAMs to M1-TAMs. Recently, we reported that sepiapterin (SEP), the endogenous $BH_4$ precursor, elevates the expression of M1-TAM markers within HER2+ tumors. Here, we show that SEP restores $BH_4$ levels in M2-like macrophages, which then redirects arginine metabolism to NO synthesis and converts M2 type to M1 type. The reprogrammed macrophages exhibit full-fledged capabilities of antigen presentation and induction of effector T cells to trigger immunogenic cell death of HER2+ cancer cells. This study substantiates the utility of SEP in the metabolic shift of the HER2+ breast tumor microenvironment as a novel immunotherapeutic strategy.**

## Introduction

Tumor-associated macrophages (TAMs) are a heterogeneous group of macrophages within solid tumors, comprising up to 50% of the cell mass of tumors (1). They largely account for the immuno-suppressive nature of the tumor microenvironment (TME) owing to the predominance of the immune-suppressive M2 type over the immune-stimulatory M1 type (2, 3). M1-TAMs are classically activated macrophages that induce pro-immunogenic anti-tumor responses within the TME. In response to pro-inflammatory stimuli, such as LPS, IFNγ, IL12, and GM-CSF, nascent (M0) macrophages are polarized to M1-TAMs and induce immune-stimulatory Th1

responses via antigen presentation and secretion of immunogenic chemokines (CXCL9, CXCL10) (4, 5). M1-TAMs also produce high levels of tumoricidal nitric oxide (NO) and reactive oxygen species, as well as pro-inflammatory cytokines (IL12, IL6, IL1β, TNFα) (6, 7, 8). On the contrary, M2-TAMs are alternatively activated macrophages that exert anti-inflammatory pro-tumor effects. Type 2 immunogenic stimuli, such as IL4, IL13, and M-CSF, trigger M2-TAM polarization, leading to the induction of immune-suppressive Th2 responses. M2-TAMs also produce a large amount of polyamines (PAs), polycations that promote secretion of anti-inflammatory cytokines, namely, IL10 and TGFβ, to facilitate tumor growth (9, 10, 11, 12).

Reduction of the M1/M2-TAM ratio, as seen in immune-suppressive tumors such as human epidermal growth factor receptor 2 (HER2)–positive breast tumors, aggravates tumor growth and therapy resistance. Conversely, the increased M1/M2-TAM ratio improves the tumor prognosis and therapeutic response (13, 14, 15). Thus, reprogramming M2-TAMs to M1-TAMs is an emerging therapeutic strategy currently at investigational stages. Nevertheless, most of such endeavors use pro-inflammatory agents (LPS, IFNγ, TNFα, CD40 agonists, and IL12) that could cause systemic toxicity in vivo and fail in clinical use (16, 17). Thus, it is essential to develop a therapeutic strategy to reprogram TAMs with little side effects on patients. Recently, metabolic modulation has emerged as a safe approach to reprogram TAMs, based on the finding that different TAM subtypes exhibit distinct metabolic profiles (18, 19). One such amenable metabolism is arginine catabolism (20). We previously reported that arginine catabolism in the breast was shifted from the immune-stimulatory NO synthesis pathway toward the immune-suppressive PA synthesis pathway during breast tumor progression, especially for HER2-positive tumors. This was largely attributed to NO synthase (NOS) dysfunction in the TME because of oxidative degradation of the essential enzyme cofactor, tetrahydrobiopterin ($BH_4$). Thus, replenishment of $BH_4$ in tumors by supplementing the endogenous precursor sepiapterin (SEP) effectively redirected arginine metabolism from PA to NO synthesis, reprogrammed M2-TAMs to M1-TAMs, and suppressed tumor cell growth (21, 22).

---

[1]Department of Cell & Cancer Biology, College of Medicine and Life Sciences, University of Toledo Health Science Campus, Toledo, OH, USA  [2]Division of Rheumatology, University of Colorado, Anschutz Medical Campus Barbara Davis Center, Aurora, CO, USA  [3]Department of Zoology and Physiology, University of Wyoming, Laramie, WY, USA  [4]MetroHealth Medical Center, Case Western Reserve University School of Medicine, Case Comprehensive Cancer Center, Cleveland, OH, USA

Correspondence: sxf494@case.edu

 

In the present study, we explored the mechanisms by which modulation of arginine metabolism could reprogram TAMs and determined the therapeutic potentials of SEP for HER2-positive breast cancer. We found that bimodal arginine metabolic pathways leading to NO versus PA synthesis are not only the consequences, but also the drivers of M1 versus M2 macrophage polarization. Activation of enzymatic pathways for NO or PA synthesis, as well as these metabolites, was essential to polarize nascent M0 macrophages to M1 versus M2 types, respectively. We further demonstrated that SEP-treated M2 macrophages not only elevated the expression of M1 macrophage markers, but also exhibited full-fledged M1-type functionalities, including the elevated capacities of antigen presentation and cytotoxic T-cell activation. When cocultured with these M1-reprogrammed macrophages and activated T cells, cancer cells underwent immunogenic cell death (ICD) indicated by the production of damage-associated molecular patterns (DAMPs) and apoptotic markers. Such strong pro-immunogenic, anti-tumor effects of SEP were verified using animal models of spontaneous or transplanted HER2-positive mammary tumors. These results strongly suggest the immunotherapeutic potentials of the SEP-mediated metabolic shift of TAMs for HER2-positive breast cancer.

# Results

## M1 and M2 macrophages are distinguished by the preferential production of NO versus PAs through differential arginine metabolism

Circulating monocytes that have entered the TME differentiate into nascent (M0) TAMs and then polarize into different subtypes based on the environmental cues (8, 23, 24, 25, 26). M1- and M2-TAMs are the two major subsets that represent the opposing ends of a spectrum in terms of morphology, metabolism, and functions (5, 11, 27, 28). We used in vitro TAM models that represent M0, M1, and M2 subtypes derived from THP-1 monocytic cells (Fig 1A) (29) and PBMCs (Fig 1B). To distinguish between different macrophage subsets, we profiled different macrophages based on the morphologies, metabolisms, and functional signatures. Phalloidin staining of filamentous actin and scanning electron microscopy (SEM) imaging of macrophages showed that the M0 type appeared as sparsely clustered spherical cells, whereas the M1 type manifested as more elongated, spindle-shaped cells. The M2 type, on the contrary, formed highly clustered and more spread morphology as reported previously (Fig 1C) (30, 31). M1 and M2 types were more quantitatively distinguished by their unique marker expression (6, 32, 33). The M1 type showed significantly higher levels of TNFα and TLR2 and the little expression of CD206. In contrast, the M2 type showed the little or low expression of TNFα and TLR2, but higher levels of CD206. Interestingly, the M0 type expressed both M1 and M2 macrophage markers, although at lower levels, attesting to their biopotency before polarization (Fig 1D–F). Furthermore, M1 versus M2 types are characterized by their differential arginine catabolism (20). The M1 type metabolizes arginine via NOS2 to produce NO for anti-tumor activities. Conversely, the M2 type metabolizes arginine by arginase 1 (Arg1) and then by ornithine decarboxylase 1 to

produce PAs for pro-tumor activities (22, 34, 35, 36). Consistently, we observed that M1 macrophages preferentially produced NO over PAs, whereas M2 macrophages preferred PAs over NO, as indicated by the differential NO/PA ratios (Fig 1G).

## SEP reprograms M2-to-M1 macrophages by downmodulating PAs but up-regulating NO synthesis

We previously reported that administration of SEP, the endogenous precursor of the NOS cofactor BH$_4$, could redirect arginine metabolism from PA to NO synthesis in the breast TME, which in turn up-regulated M1 macrophage marker expression while downmodulating M2 macrophage markers (22). In the present study, we sought to investigate whether the M2 type treated with SEP would indeed acquire M1-type functionalities using macrophages derived from THP-1 cells. We polarized these macrophages to M1 versus M2 types and treated them with SEP (100 $\mu$M), in comparison with the equal volume of DMSO as a vehicle control, for 3 d, and determined their phenotypic profiles. The M1 type treated with SEP (100 $\mu$M) and the M2 type treated with LPS (5 ng/ml) plus IFNγ (20 ng/ml) were used as positive controls.

First, to determine the bioactivity of SEP, we measured BH$_4$ production. Although the endogenous BH$_4$ levels were significantly lower in the M2 type than the M1 type, SEP treatment elevated BH$_4$ in both M1 and M2 types to almost equal levels (Fig 2A). The SEP-treated M2 type showed M1 macrophage–like morphology characterized by isolated spindle shapes (Fig 2B). Consistent with our previous report (22), the SEP-treated M2 type showed large increases in M1-type markers (TLR2, TNFα, IL1β, and IL6) and decreases in M2-type markers (CD206 and TGFβ), although to a lesser degree than a positive control (LPS plus IFNγ) (Fig 2C–F). We then tested the effects of SEP on arginine metabolism of macrophages. Consistent with our previous findings (22), the M1 type produced threefold higher levels of NO than the M2 type, whereas the M2 type produced fourfold higher levels of PAs than the M1 type. Thus, the NO/PA ratio of the M1 type was over eightfold higher than that of the M2 type. Upon SEP treatment, however, the NO/PA ratio in the M2 type increased by threefold, whereas it did not change in the M1 type (Fig 3A). These results suggested that SEP redirected the arginine metabolism of the M2 type from PA to NO synthesis.

To explore how SEP-mediated modulation of arginine metabolism had influenced macrophage polarization, we determined the relevance of NO versus PA synthesis pathways to M1 versus M2 macrophage polarization, respectively. We inhibited the NO synthesis pathway with a specific NOS2 inhibitor 1400W (50 $\mu$M), while inhibiting the PA synthesis pathway with a specific ARG1 inhibitor nor-NOHA (50 $\mu$M). As expected, 1400W significantly downmodulated NO levels in the M1 type, whereas nor-NOHA downmodulated PA levels in the M2 type (Fig 3B). Concomitantly, the 1400W-treated M1 type showed a decrease in M1 macrophage markers, TNFα and IL12, but an increase in M2 macrophage markers, CD206 and IL10. In contrast, the nor-NOHA–treated M2 type showed decreases in M2 macrophage markers, but increases in M1 macrophage markers (Fig 3C and D). These results suggest that NOS2 versus ARG1 functions are essential for the polarization to M1 versus M2 macrophages, respectively (Fig 3E). We further tested the relevance of NO versus PA per se to M1 versus M2 macrophage

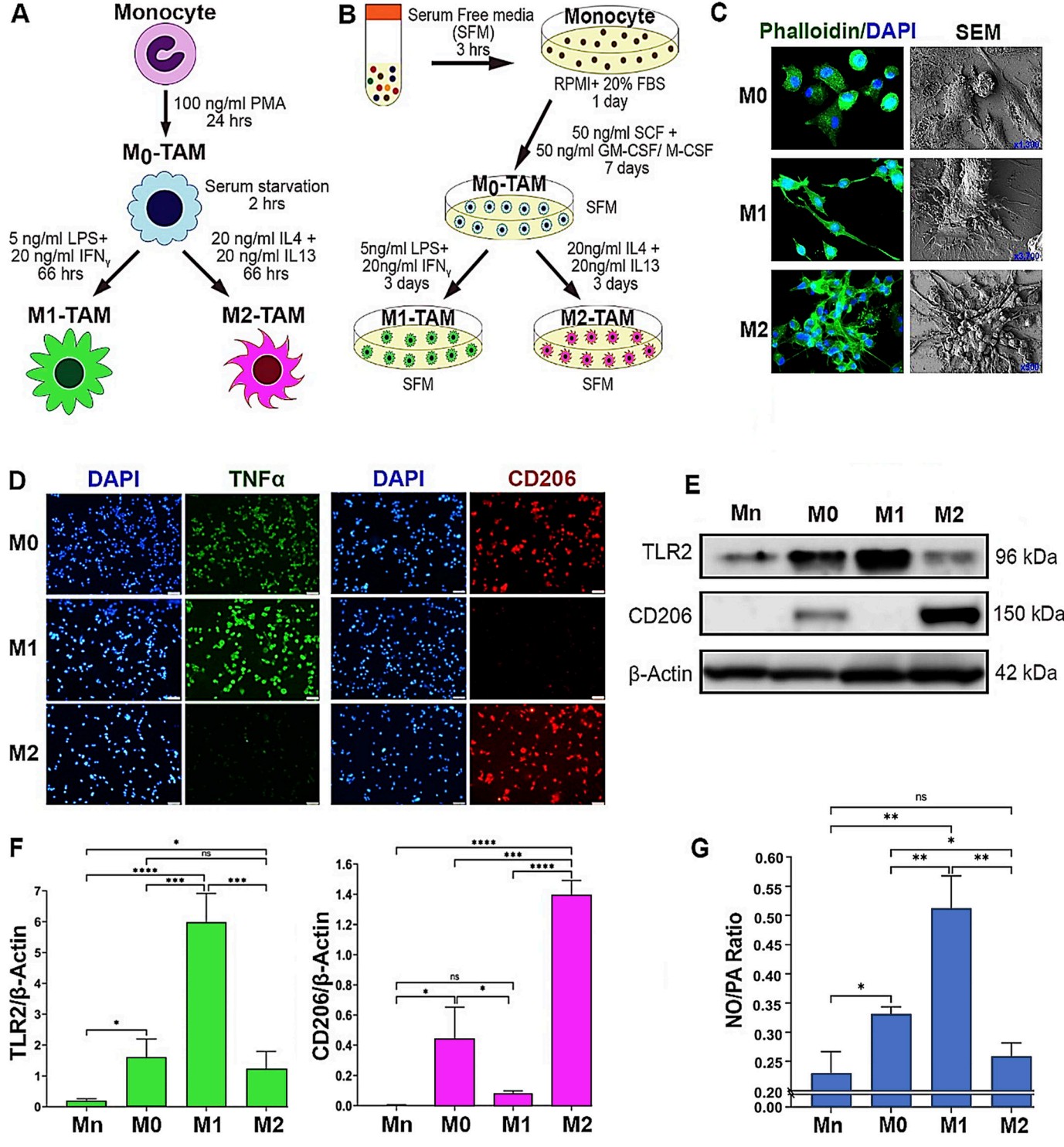

**Figure 1. M1- versus M2-TAMs are distinguished by the preferential production of NO versus PAs through differential arginine metabolism.**
**(A, B)** Schematic representation of the polarization protocols of TAMs derived from the THP-1 human monocytic cell line (A) and PBMC (B). THP-1 monocyte (Mn) was treated with phorbol myristate acetate for differentiation into inactive macrophages ($M_0$). PBMC-derived Mn was treated with SCF+GM-CSF (leading to M1) or M-CSF (leading to M2) for differentiation into $M_0$. For M1 polarization, $M_0$-TAMs were treated with LPS and IFNγ; for M2 polarization, $M_0$-TAMs were treated with IL4 plus IL13. **(C)** Representative images of phalloidin staining (left, n = 3) and scanning electron microscopy (SEM, right, n = 3) imaging of M0-, M1-, and M2-TAMs. Green: phalloidin; blue: DAPI. SEM images are shown at different magnifications: M0: x1,300; M1: x3700; and M2: x500. **(D)** Immunofluorescence images of TAM subsets (n = 3) stained for M1 (green, TNFα) versus M2 markers (red, CD206) and counterstained with DAPI (blue). Scale bars: 50 $\mu$m. **(E)** Western blot analysis (n = 3) of THP-1–derived Mn-, M0-, M1-, and M2-TAMs for the expression of TLR2 (M1 marker) versus CD206 (M2 marker). β-Actin was used as the internal loading control. **(F)** Quantification of the Western blot

polarization (22), respectively. Although we and others observed that NO and PAs were produced as the consequence of M1 versus M2 polarization (20), we suspected that both metabolites could also be the "driver" of M1 versus M2 polarization. First, we treated nascent M0 macrophages with SEP, NO donors (GSNO and SNAP), or PA (spermine) and measured M1 and M2 marker expression. M0 macrophages treated with SEP or NO donors significantly elevated M1 markers (TLR2, STAT1, pSTAT1$_{S727}$, and IL12), while decreasing M2 markers (CD163, STAT3, and IL10) (Figs 4A–C and S1A and B). Next, we treated M1 macrophages with NO scavenger cPTIO and M2 macrophages with an inhibitory PA analog DENSPM. We showed that cPTIO effectively inhibited M1 markers, whereas DENSPM inhibited M2 markers (Figs 4D–F and S2A and B). These results suggest that NO and PAs are indeed drivers of M1 versus M2 macrophage polarization, accounting for the mechanism of SEP-induced reprogramming of M2 to M1 types (Fig 4G).

## SEP-treated M2 TAMs exhibit enhanced antigen presentation capabilities

To validate that these reprogrammed macrophages are indeed functional, we measured their antigen presentation capabilities. Macrophages are professional APC that activate the adaptive immune system upon detection of foreign antigens (37, 38, 39). M1 macrophages could present neoantigens derived from tumor cells toward CD4$^+$ type I T helper cells (Th1) via HLA-DR (MHC II) receptors, which in turn activate CD8$^+$ cytotoxic T cells (40, 41, 42). M2 macrophages, on the contrary, suppress cytotoxic T-cell activity by expressing the tolerogenic HLA-G antigen (13, 43). To measure antigen presentation capability, we used an OVA $_{323–339}$ (OVA) peptide, an antigenic epitope of HLA-DR (44, 45) that would be presented via the cell surface HLA-DR (MHC II) on M1 macrophages toward the TCR on Th1 cells (Fig 5A). We then measured the levels of the cell surface HLA-DR, which would emerge after antigen binding. As expected, in the absence of OVA, only the basal level of surface HLA-DR was detected for all macrophage subsets. Upon being pulsed with OVA, however, the M1 type expressed significantly higher levels of the cell surface HLA-DR than the M2 type. Conversely, SEP treatment dramatically elevated the cell surface HLA-DR levels in the M2 type (fivefold) and the M1 type (1.5-fold). Overall, these findings demonstrate that SEP significantly elevated the antigen presentation capability of the M2 type to the levels equivalent to that of the M1 type (Fig 5B–D).

## SEP-treated M2 macrophages effectively activate cytotoxic T cells

We next tested whether these reprogrammed macrophages could indeed activate cytotoxic T cells. Cytotoxic T cells are the major component of the adaptive immune system and the executors of anti-tumor immune responses. Cytotoxic T cells are activated through antigen presentation and induce cancer cell death by releasing cytotoxic proteins such as granzymes, perforin, and IFNγ (46, 47, 48, 49). M1 macrophages are able to activate cytotoxic T cells,

whereas M2 macrophages instead inhibit their activation (41). To determine T-cell activation, we measured their IFNγ production, cell proliferation, and CD107a surface expression (degranulation) (50, 51, 52, 53) (Fig 6A). To this end, we devised two different co-culture systems: indirect (Transwell-based) and direct methods, to co-culture macrophages, T cells, and breast cancer cells. The indirect method was used to co-culture THP-1–derived macrophages, PBMC-derived T cells, and cancer cells, whereas the direct method was used to co-culture PBMC-derived macrophages and T cells along with cancer cells (Fig 6B). Here, we specifically focused on HER2+ breast cancer cells based on our previous studies demonstrating the pathogenesis of this cancer subtype responsive to dysregulated NO levels and their great sensitivity to SEP treatment (21, 22, 54). We observed that in these triple co-cultures, M1 macrophages, but not the M2 type, strongly induced T cells to produce IFNγ, proliferate, and express cell surface CD107a (Fig 6C–G). However, SEP-treated M2 macrophages dramatically elevated these three activation markers in co-cultured T cells. SEP-treated M1 macrophages, however, did not exhibit a further increase in T-cell activation, indicating the existence of certain threshold levels. These results strongly suggest that M2 macrophages treated with SEP could strongly activate cytotoxic T cells, as the result of their functional reprogramming to M1 macrophages.

## SEP-treated M2 macrophages induce ICD in HER2-positive breast cancer cells

Cytotoxic activities of T cells could be determined by measuring the death of target cells. To validate the cytotoxic activity of T cells co-cultured with macrophages, HER2+ breast cancer cells (BT474 and SKBR3) and the conditioned media (CM) from co-cultures were analyzed for cell death markers. Cell cycle profiling demonstrated that cancer cells co-cultured with SEP-treated M2 macrophages along with T cells showed a great increase in the sub-G1 (apoptotic) population compared with those co-cultured with control M2 macrophages (Figs 7A and B and S3A and B). This observation was further confirmed by the large increases in cancer cells positive for Annexin V (total apoptotic cells) at both early (PI low) and late (PI high) stages of apoptosis after being co-cultured with SEP-treated M2 macrophages and T cells (Fig 7C–E).

To determine the mechanisms of cancer cell death after being co-cultured with SEP-treated macrophages along with T cells, we analyzed the CM for the levels of secreted ATP, a type of DAMPs released from cells undergoing ICD. DAMPs bind and activate the cognate cell surface receptors on phagocytic cells to mediate their own destruction (55, 56). CM of T cell–only cultures contained the basal levels of ATP regardless of the treatment. The inclusion of cancer cells, however, increased the extracellular ATP levels by twofold, which were increased by SEP treatment by an additional 20%. The further addition of M1 macrophages, but not the M0 or M2 type, increased the secreted ATP levels by another 50%. Nevertheless, when M2 macrophages were pretreated with SEP, the addition of this M2 type elevated the extracellular ATP to the levels

---

results of the expression of TLR2 (left) and CD206 (right) normalized against β-actin and presented as fold differences. **(G)** NO-to-PA ratios in THP-1–derived TAM subsets (n = 6) measured with ELISA. Error bars: ±SEM. *$P \le 0.05$; **$P \le 0.01$; ***$P \le 0.001$; ****$P \le 0.0001$; and ns, $P > 0.05$.

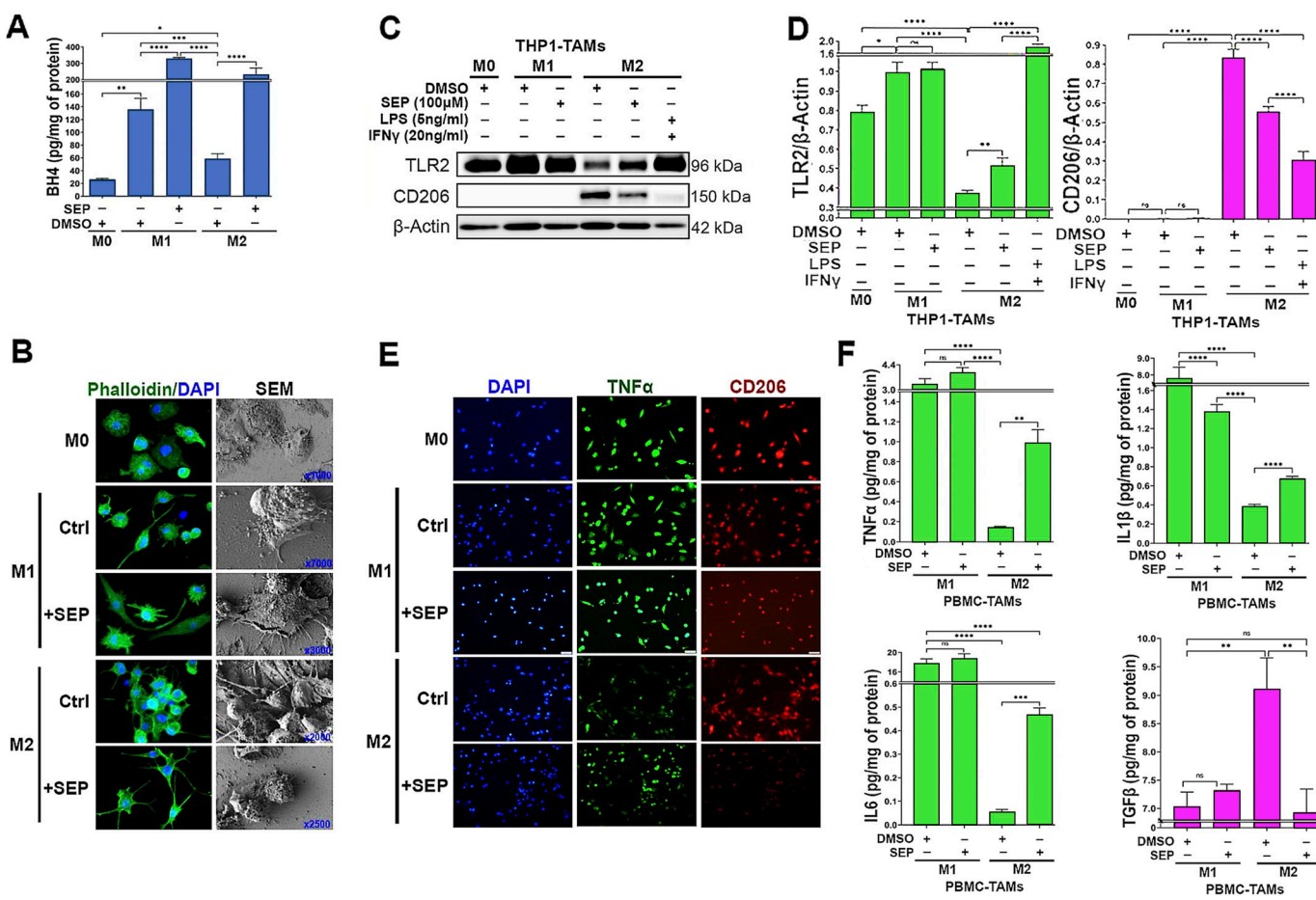

**Figure 2. SEP elevates M1 marker expression in M2-TAMs.**
**(A)** Levels of BH$_4$ produced by THP–1–derived M0-, M1-, and M2-TAMs after being treated with vehicle (DMSO) or SEP (100 $\mu$M) for 3 d (n = 6). BH$_4$ levels were measured with ELISA and normalized against the total protein levels. One-way ANOVA with a post hoc test (Tukey's test) was performed to measure the significance of the mean difference between treatment groups. **(A, B)** Phalloidin staining and SEM imaging of THP-1–derived M0-, M1-, and M2-TAMs treated as in (A) (n = 3). SEM images are shown at different magnifications. **(A, C)** Western blot analysis of TAM subsets treated with a vehicle or SEP as in (A), and $\beta$–actin was used as the internal loading control (n = 5). **(D)** Quantification of the Western blot results based on the expression of TLR2 (M1 marker) versus CD206 (M2 marker) normalized against $\beta$–actin signal and presented as fold differences. **(E)** Immunofluorescence imaging of THP-1–derived TAMs after treatments as shown above, and stained for an M1 marker (green, TNF$\alpha$) versus M2 marker (red, CD206) and counterstained with DAPI (blue) (n = 3). **(F)** Levels of secreted cytokines, type 1: TNF$\alpha$ (top left), IL1$\beta$ (top right), and IL6 (bottom left) versus type 2: TGF$\beta$ (bottom right), for M1- versus M2-TAMs treated with a vehicle versus SEP (n = 6) measured with ELISA. Error bars: ±SEM. *$P \leq 0.05$; **$P \leq 0.01$; ***$P \leq 0.001$; ****$P \leq 0.0001$; and ns, $P > 0.05$.

equivalent to those of co-cultures with the M1 type (Fig 7F). These findings demonstrated that SEP greatly enhanced cancer cell–killing activities of adaptive immunity through ICD.

### Oral SEP treatment elevates M1 macrophages in TME and suppresses the growth of spontaneous MMTV-neu (HER2) mammary tumors

To validate the therapeutic efficacy of SEP, we gave SEP to MMTV-neu mice, which were a mouse model of spontaneous HER2-positive mammary tumors. These animals developed single-focal tumors at the latencies of 6–14 mo. Once tumors became palpable, animals were divided into the control (DMSO) versus SEP (10 mg/kg) treatment groups and given the drug through ad libitum access to acidified drinking water for 6 wk (Fig 8A). Tumor growth was measured twice a week using a caliper, and the morbidity of

animals was also observed. We saw about 50% reduction in the tumor growth curve, as well as the sizes of the excised tumors of the SEP-treated group with statistical significance (Fig 8B and C). We did not see any morbidity in animals because of treatments. To determine the immunogenicity of these tumors, we isolated TAMs and compared their M1- versus M2-TAM marker expression. In the SEP-treated group, we saw large increases in M1 markers (CD80, IL12, and IFN$\gamma$) but a decrease in an M2 marker, CD163. Another M2 marker, IL10, however, was not influenced by SEP treatment, indicating the persistence of a subtype of M2-TAMs (Fig 8D and E). Still, a large increase of M1-TAMs in SEP-treated tumors is expected to be the significant contributor to the tumor inhibitory effects of the drug. Such increase in M1-TAMs and decrease in M2-TAMs by SEP treatment were more consistent throughout different markers in tumors than spleens, indicating that the major targets of SEP were TAMs (Fig 8E). Our results altogether suggest that strong tumor-

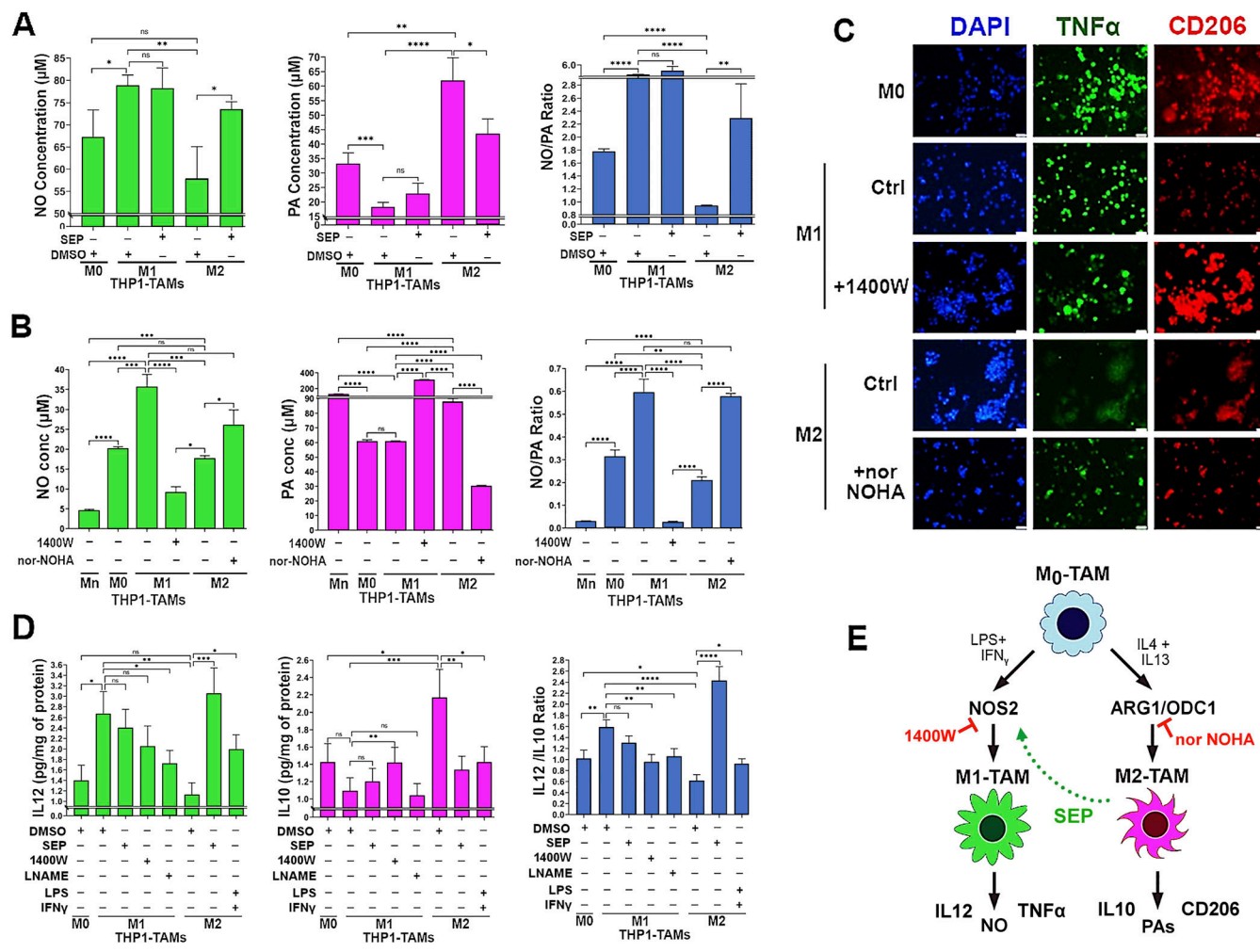

**Figure 3. SEP redirects arginine metabolism from PA to NO synthesis in M2-TAMs, while rendering them an M1-TAM phenotype.**
**(A)** Levels of NO (left), PAs (middle), and NO/PA ratios for THP–1–derived M0-, M1-, and M2-TAMs after being treated with DMSO (vehicle) and SEP (100 $\mu$M) for 3 d (n = 5).
**(B)** One-way ANOVA with post hoc Tukey's test was used for statistical analysis. Error bars: ±SEM. GraphPad Prism version 9.5.1. was used to perform all statistical analyses.
**(B)** Levels of NO (left), PAs (middle), and NO/PA ratios for THP-1–derived Mn-, M0-, M1-, and M2-TAMs after being treated with DMSO (vehicle), NOS2 inhibitor, 1400W (50 $\mu$M), or arginase 1 (Arg1) inhibitor, nor-NOHA (50 $\mu$M), for 3 d (n = 5). Note the significant decrease of the NO level in 1400W-treated M1-TAMs and the significant decrease of the PA level in nor-NOHA–treated M2-TAMs. **(C)** Immunofluorescence imaging of THP-1–derived M0-, M1-, and M2-TAMs stained for an M1 marker (green, TNF$\alpha$) versus M2 marker (red, CD206) and counterstained with DAPI (blue). M1-TAMs were treated with DMSO (control: Ctrl) or NOS2 inhibitor (100 $\mu$M 1400W), whereas M2-TAMs were treated with DMSO (Ctrl) or ARG1 inhibitor (50 $\mu$M nor-NOHA) for 3 d (n = 3). **(D)** Levels of type 1 cytokine IL12 (left) and type 2 cytokine IL10 (middle), as well as IL12/IL10 ratios for THP-1–derived TAM subsets measured with ELISA. M1-TAMs were treated with DMSO or NOS inhibitors, 1400W (50 $\mu$M) and L-NAME (2.5 mM). M2-TAMs were treated with DMSO, SEP (100 $\mu$M), or positive control LPS (5 ng/ml) plus IFN$\gamma$ (20 ng/ml) for 3 d (n = 6). The cytokine levels were measured using ELISA and normalized against the total protein levels. Error bars: ±SEM. *$P \le 0.05$; **$P \le 0.01$; ***$P \le 0.001$; ****$P \le 0.0001$; and ns, $P > 0.05$. **(E)** Working scheme for the induction of M1 versus M2 polarization by activation of NOS2 versus ARG1/OCD1 pathways and M2-to-M1 reprogramming by SEP.

suppressive effects of SEP are linked to the immunogenic shift of TAMs from M2 to M1 types.

## Suppressive effects of oral SEP treatment on mice with HER2 tumors are linked to the elevation of anti-tumor lymphocytes in the circulation and within tumors

To confirm that SEP-mediated tumor suppression was indeed attributed to its immune-stimulatory effects, we determined the impact of SEP treatment on the levels of anti-tumor lymphocytes, in particular, cytotoxic CD8⁺ T cells. We orthotopically injected NT2.5 cells, derived from MMTV-neu mammary tumors (57), into immunocompetent FVB/NJ

mice and treated these mice with DMSO or SEP (10 mg/kg) in drinking water for 5 wk. Consistent with our previous experiment (Fig 8), SEP treatment significantly suppressed tumor growth (Fig 9A and B). Interestingly, these tumors contained mammary gland–like structures composed of well-differentiated epithelial cells with heterochromatic nuclei. In contrast, DMSO-treated tumors were mostly composed of highly proliferative parenchyma with euchromatic nuclei (Fig 9C). Importantly, SEP-treated tumors showed abundant M1 macrophages and CD8⁺ T cells, indicating the highly immunogenic TME. This was in stark contrast to DMSO-treated tumors that contained prominent M2 macrophages and CD4⁺ and FOXO3+ Treg cells, indicating the immune-suppressive TME (Fig 9D).

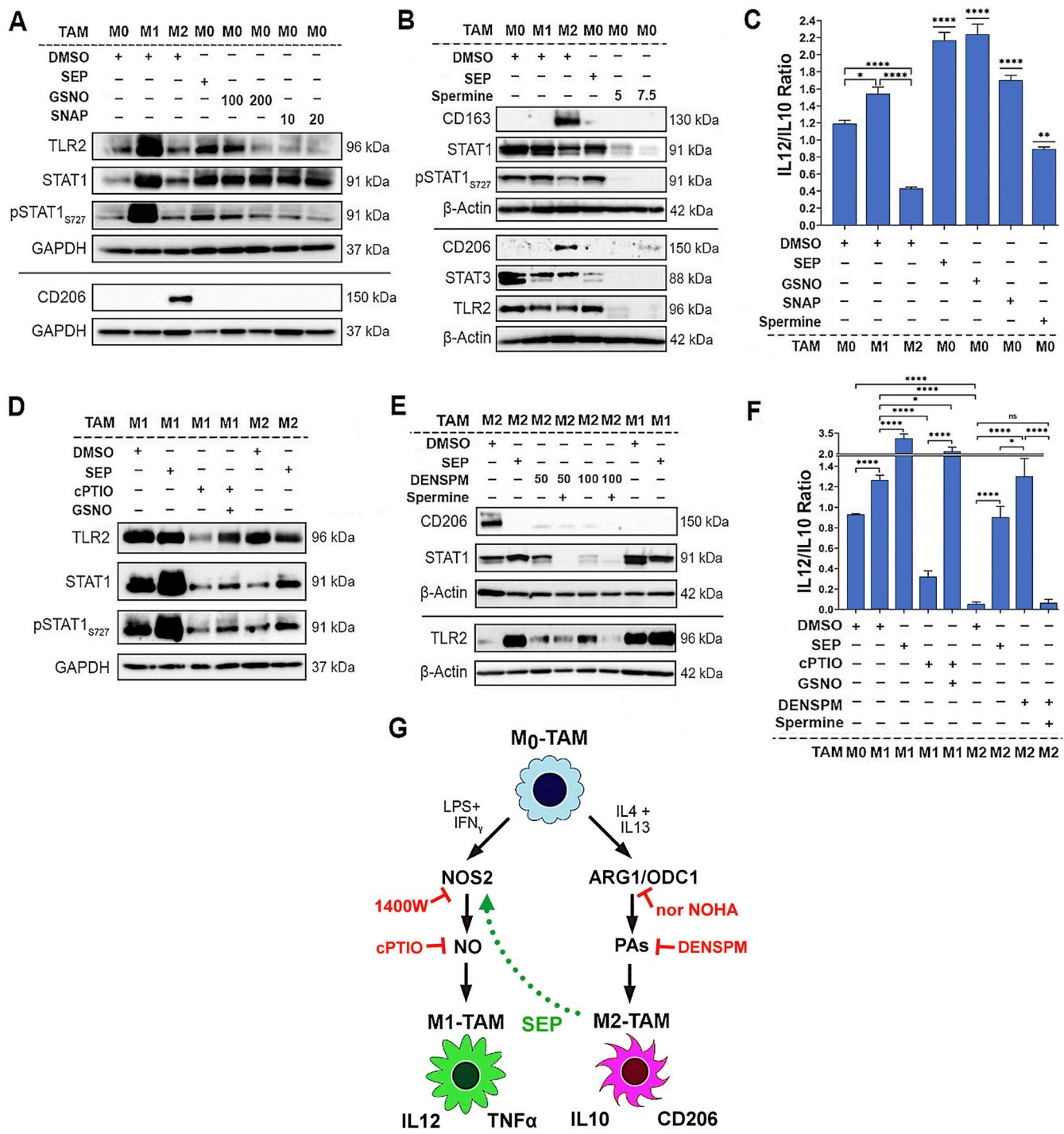

**Figure 4. Arginine metabolites, nitric oxide and polyamines, drive M1- and M2-TAM polarization, respectively.**
**(A)** Western blot analysis on M1 markers: TLR2, STAT1, and pSTAT1$_{S727}$ versus M2 marker: CD206 in THP-1–derived M0-TAMs treated with DMSO (vehicle control), SEP (100 $\mu$M), or NO donor [GSNO (100 and 200 $\mu$M) and SNAP (10 and 20 $\mu$M)] in comparison with M1- and M2-TAMs (n = 4). GAPDH was used as the internal loading control.
**(B)** Western blot analysis on M1 markers: STAT1 and TLR2 versus M2 marker: CD206 in M0-TAMs treated with DMSO (vehicle control), SEP (100 $\mu$M), or PAs (5 or 7.5 mM spermine) in comparison with M1- and M2-TAMs (n = 4). **(A, B)** $\beta$-Actin was used as the internal loading control. (For quantification of (A, B), see Appendix Fig S1.) **(C)** Ratios of IL12 to IL10 secreted by THP-1–derived M0-TAMs treated with DMSO, SEP, NO donors, and PAs in comparison with M1- and M2-TAMs. **(D)** Western blot analysis on M1 markers: TLR2, STAT1, and pSTAT1$_{S727}$ in M1-TAMs treated with NO scavenger (50 $\mu$M cPTIO) with and without NO donor (100 $\mu$M GSNO) and M2-TAMs treated with DMSO or SEP (n = 4). GAPDH was used as the internal loading control. **(E)** Western blot analysis on M1 markers: TLR2, STAT1, and pSTAT1$_{S727}$ versus M2 marker: CD206 in THP-1–derived M2-TAMs treated with a PA analog (50 and 100 $\mu$M DENSPM) and PAs (5 mM spermine) and M1-TAMs treated with DMSO or SEP (n = 4). **(D, E)** $\beta$-Actin was used as the internal loading control. (For quantification of (D, E), see Appendix Fig S2.) **(D, E, F)** Ratios of IL12 to IL10 secreted by THP-1–derived M1- and M2-TAMs with treatment combinations shown in (D, E). Error bars: ± SEM. GraphPad Prism version 9.5.1. was used to perform all statistical analyses. *$P \leq 0.05$; **$P \leq 0.01$; ***$P \leq 0.001$; ****$P \leq 0.0001$; and ns, $P > 0.05$. **(G)** Scheme for the induction of M1 versus M2 polarization by NO versus PAs and M2-to-M1 reprogramming by SEP.

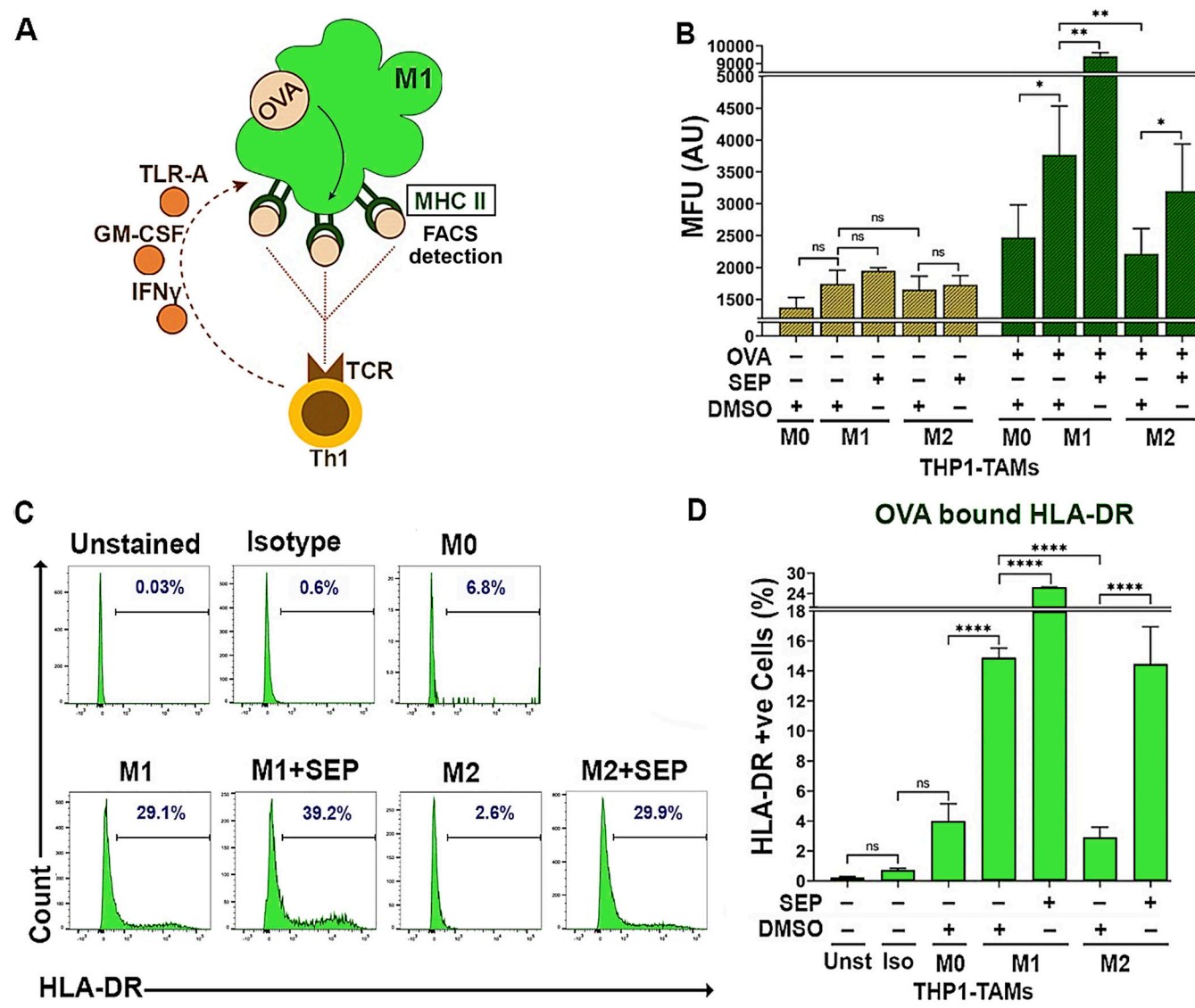

**Figure 5. M2-TAMs treated with SEP show increased antigen presentation activities.**
**(A)** Scheme of measuring antigen presentation activities of TAMs. Once the M1 macrophage is pulsed with an OVA$_{323-339}$ peptide, it phagocytoses and presents the epitope through the cell surface MHC II. The level of cell surface MHC II, representing antigen presentation activity, is detected by FACS. (The presented epitope is then recognized by TCR on Th1 T cells to trigger immunogenic responses.) **(B)** Mean fluorescence intensity of cell surface HLA-DR (MHC II) after being pulsed with or without an OVA$_{323-339}$ peptide (20 µg/ml for 2 h) on THP-1–derived $M_0$-, M1-, and M2-TAMs pretreated with DMSO or 100 µM SEP. Two-sample $t$ tests (unpaired) were performed for pairwise comparison. **(B, C, D)** Percentages of TAM subsets treated as in (B) that expressed cell surface HLA-DR (bound by an OVA peptide) presented as histograms (C) and quantification (D). Unstained (Unst) and isotype (Iso) controls were used (n = 5). Error bars: ± SEM. *$P \leq 0.05$; **$P \leq 0.01$; ***$P \leq 0.001$; ****$P \leq 0.0001$; and ns, $P > 0.05$.

Complementarily, we tested whether SEP would exert tumor-preventive effects by influencing the systemic immunity. Here, we employed MMTV-neu/FVB mice at the prepubertal stage (4 wk old), which had not developed mammary tumors. We treated these mice with DMSO or SEP (1 mg/kg) in drinking water for 8 mo to monitor tumor incidence and immune cell compositions in the blood. As expected, SEP treatment significantly improved tumor-free survival of MMTV-neu mice, with the delay of 50% tumor incidence by over 50 d (Fig 9E). Next, we sought to profile immune cells in the blood by single-cell sequencing. For this analysis, we grouped these animals into four groups: DMSO with (80% of the DMSO group) or without

tumors (20% of the DMSO group); and SEP with (45% of the SEP group) or without tumors (55% of the SEP group). We isolated PBMCs from the four groups and compared their major cell types (Fig 9F). Surprisingly, the SEP without tumor group was highly abundant in T lymphocytes primarily composed of natural killer cells, and effector CD8[+] and CD4[+] T cells. The DMSO without tumor group was instead abundant in B lymphocytes, further suggesting the roles of anti-tumor lymphocytes in tumor prevention. In contrast, the DMSO with tumor group was abundant in granulocyte populations largely composed of neutrophils, whereas the SEP with tumor group was abundant in macrophages, which were presumably pro-tumor type

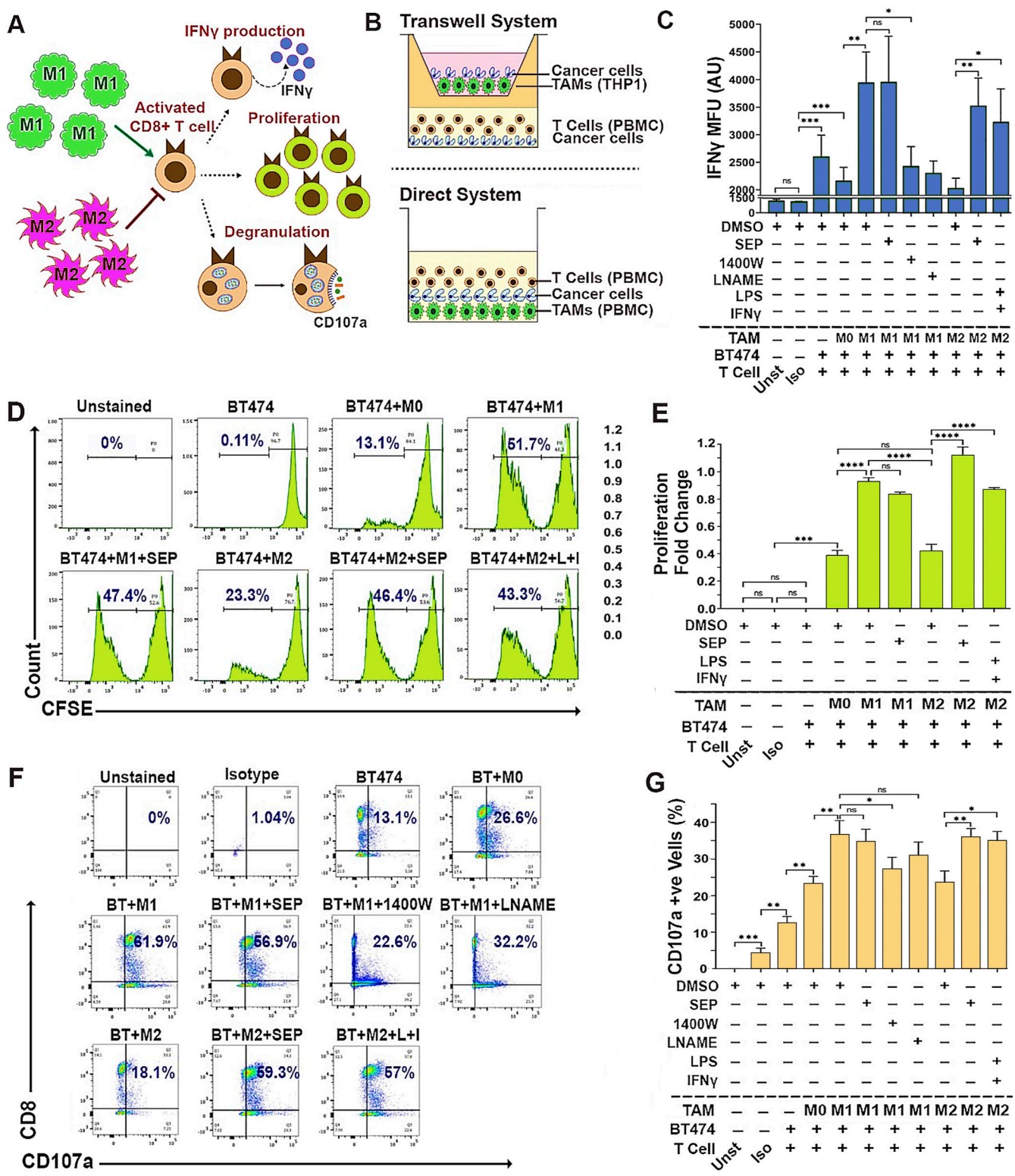

**Figure 6. SEP-treated M2-TAMs activate cytotoxic T cells.**
**(A)** Scheme of detection methods for activation of cytotoxic CD8[+] T cells, based on IFNγ production, proliferation, and degranulation indicated by cell surface CD107a expression. **(B)** Schemes of the TAM–T-cell–cancer cell co-cultures (2:2:1 ratio) using the Transwell system (top) and direct system (bottom). (Top) THP-1–derived TAMs, PBMC-derived T cells, and BT474 breast cancer cells co-cultured using Transwell. (Bottom) PBMC-derived autologous TAMs and T cells were directly co-cultured with BT474 cells. **(C)** FACS-detected IFNγ expression levels in CD8[+] T cells after being co-cultured with BT474 cells and PBMC-derived TAM subsets pretreated with DMSO or 100 μM SEP (n = 6). Positive control: M2-TAMs treated with LPS and IFNγ. Negative control: M1-TAMs treated with 1400W and L-NAME. T cells were gated for CD3 and CD8 expression.

(Fig 9G). These results confirmed that anti-tumor effects of SEP were strongly linked to the elevated levels of anti-tumor T lymphocytes in circulation and within tumors in association with the immunogenic shift of tumor-associated macrophages. Our results altogether strongly suggest the potential of SEP as a novel immunotherapeutic agent for HER2-positive breast cancer.

## Discussion

TAMs are a group of heterogeneous myeloid cells that manifest great plasticity under the influence of varying signals from the TME. Although TAMs are broadly classified into the pro-inflammatory M1 versus anti-inflammatory M2 types for simplicity's sake, their given phenotypes could range within the spectrum. Both types of TAMs encompass distinct metabolic profiles depending on their specific needs for energy production. M1 TAMs tend to be localized in hypoxic environments that trigger glycolytic signaling. Thus, they primarily use glycolysis for energy production, whereas mitochondrial oxidative phosphorylation, namely, tricarboxylic acid cycle and electron transport chain, is largely compromised. M1 TAMs also up-regulate lipid biosynthesis necessary for executing inflammatory responses. On the contrary, M2-TAMs lack a sufficient supply of glucose for glycolysis because of the high glucose consumption by tumor cells. Thus, they resort to mitochondrial oxidative phosphorylation and fatty acid oxidation as the major sources of energy production (58).

Interestingly, recent studies have unraveled that these distinct metabolic propensities of M1- versus M2-TAMs are largely contributed by their differential arginine metabolism. In M1-TAMs that up-regulate NOS2, arginine is primarily converted to NO. NO is found to induce S-nitrosylation (conjugation of NO to a cysteine residue) and inhibition of several iron–sulfur proteins present in the mitochondrial electron transport chain, such as complex I and cytochrome c oxidase. This renders M1-TAMs heavily dependent on glycolysis for energy production (59). Conversely, in M2-TAMs that up-regulate arginase/ornithine decarboxylase, arginine is preferentially converted to ornithine and then PAs. One of such PAs, spermidine, is found to induce hypusination (conjugation of spermidine to a lysine residue) and activation of eukaryotic initiation factor 5A, the protein responsible for the translation of several enzymes involved in the mitochondrial tricarboxylic acid cycle, such as succinyl-CoA synthetase, succinate dehydrogenase, methylmalonyl-CoA mutase, and pyruvate dehydrogenase. This allows M2-TAMs to resort to mitochondrial oxidative phosphorylation for energy production when there is not enough supply of glucose for glycolysis (60).

Given that distinct arginine metabolites help induce the formation of different TAM types, we hypothesized that a shift of arginine metabolism might trigger TAM reprogramming. In the present study, we tested whether supplementing SEP, the endogenous precursor of the NOS cofactor $BH_4$, would redirect arginine metabolism toward NO synthesis and induce reprogramming of M2-TAMs to M1-TAMs. This was based on the previous findings of ours and others that $BH_4$ biosynthesis is activated in response to the stimuli involved in M1-TAM polarization (namely, $TNF\alpha$, $IFN\gamma$, and LPS), but is downmodulated in M2-TAMs (22, 61). SEP is a cell-permeable pteridine originally discovered as a yellow eye pigment of *Drosophila melanogaster* (62). SEP is ubiquitously synthesized from bacteria to mammals as part of the salvage pathway of $BH_4$ biosynthesis (63). Administering SEP is found to be much more efficient in elevating the intracellular $BH_4$ levels than administering $BH_4$ itself (64, 65). SEP has been tested as a therapeutic for different metabolic disorders, such as diabetes, hypertension, and cardiovascular diseases (66, 67). It has also been used for treating phenylketonuria (PKU), the condition defective in phenylalanine catabolism, in phase III clinical trials (68, 69, 70, 71). We recently reported that supplementing SEP could shift arginine metabolism from PA to NO synthesis in both HER2-positive breast cancer cells and macrophages, inhibiting the growth of mammary tumors (21, 22). In the present study, we further explored how SEP induces reprogramming of M2- to M1-TAMs, as well as its therapeutic efficacies for promoting T cell–mediated anti-cancer immunity. We showed here that the addition of SEP alone was sufficient to elevate NO-to-PA ratios in both nascent M0 macrophages and immune-suppressive M2 macrophages and polarize them toward the M1 type. These repolarized macrophages manifested bona fide M1-type functions, including the full-fledged capabilities of antigen presentation, T effector cell activation, and induction of ICD of HER2-positive tumor cells. We confirmed the pro-immunogenic, anti-tumor effects of SEP using an animal model of spontaneous and transplanted HER2-positive mammary tumors.

Although the present study specifically focuses on the utility of SEP in elevating the immunogenicity of TAMs in HER2-positive breast cancer, the SEP/$BH_4$ pathway could also be used for the activation or protection of other cell types. For example, the SEP/$BH_4$ pathway is critically involved not only in T-cell proliferation for anti-tumor activities (72), but also in protection of vascular functions in response to inflammation-induced endothelial damages (61). Furthermore, because $BH_4$ serves as the cofactor of several different enzymes (NOS, phenylalanine/tyrosine/tryptophan hydroxylases, and alkylglycerol monooxygenase), it could exert additional beneficial effects on energy metabolism, redox mechanisms, and disease treatments independent of NOS (73, 74). Besides, both $BH_4$ and SEP are FDA-approved drugs (sold commercially under the names sapropterin and PTC923, respectively), and their "off-label uses" have tremendously grown during the last two decades through a number of clinical trials for various conditions including cardiac, pulmonary, rheumatologic, dermic, and psychiatric diseases,

---

IFNγ levels are shown as MFU. **(C, D)** Detection of proliferation of cytotoxic T cells co-cultured as in (C) based on the dilution of CFSE signals through cell doubling (n = 6). Percentages of proliferating T cells (CFSE low) are highlighted in the histogram. **(E)** Cytotoxic T-cell proliferation shown as fold change of proliferating cells (CFSE low) with respect to non-proliferating cells (CFSE high). **(B, F)** Cell surface CD107a (degranulation marker) expression on cytotoxic T cells directly co-cultured as in (B). Percentages of CD107a+ CD8$^+$ cells are shown in the plots. **(G)** Quantification of the percentages of CD107a+ CD8$^+$ cells in co-cultures. Error bars: ± SEM. *$P$ ≤ 0.05; **$P$ ≤ 0.01; ***$P$ ≤ 0.001; ****$P$ ≤ 0.0001; and ns, $P$ > 0.05.

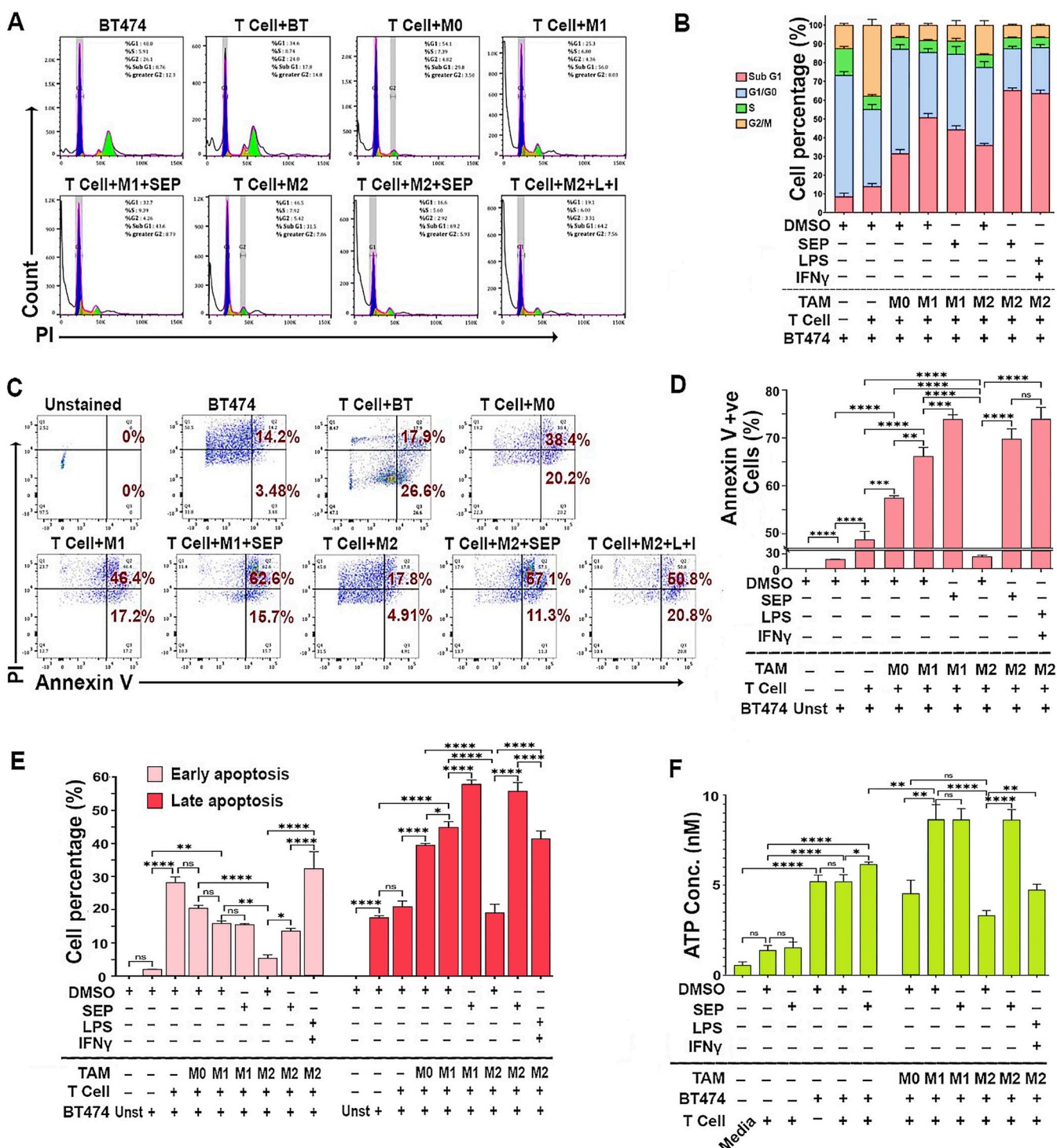

**Figure 7. SEP-treated M2-TAMs induce T cells to kill HER2+ breast cancer cells.**
**(A)** Cell cycle analyses of BT474 cancer cells (CMFDA-labeled) co-cultured with PBMC-derived TAM subsets, pretreated with DMSO (vehicle), SEP (100 μM), or LPS + IFNγ (positive control), along with T cells. Adherent cells (TAMs+BT474 cells) were dissociated, fixed in 70% ethanol for 3 h, and stained with PI. BT474 cells were gated based on the CMFDA signal and analyzed for the PI-stained DNA contents. **(A, B)** Cell cycle distribution of BT474 cells co-cultured with TAMs and T cells as in (A). Note the dramatic increase in sub-G1 population co-cultured with SEP-treated M2-TAMs. (For quantification of sub-G1 and G1/G0 populations, see Appendix Fig S3A). **(C)** Annexin V/PI staining of co-cultured BT474 cells to measure cell deaths (n = 6). Viable cells: Annexin V -ve, PI -ve; early apoptotic cells: Annexin V +ve, PI -ve; late apoptotic cells: Annexin V +ve, PI +ve; and necrotic cells: Annexin V -ve, PI +ve. **(D)** Percentage of total apoptotic (Annexin V +ve) cancer cells. **(E)** Early and late apoptotic cancer cells. **(F)** Levels of ATP secreted by BT474 cells in co-cultures. Secretion of ATP indicates immunogenic cell death of cancer cells. Note the dramatic increase in ATP secretion by cancer cells in co-culture with SEP-pretreated M2-TAMs. Error bars: ± SEM. *P ≤ 0.05; **P ≤ 0.01; ***P ≤ 0.001; ****P ≤ 0.0001; and ns, P > 0.05.

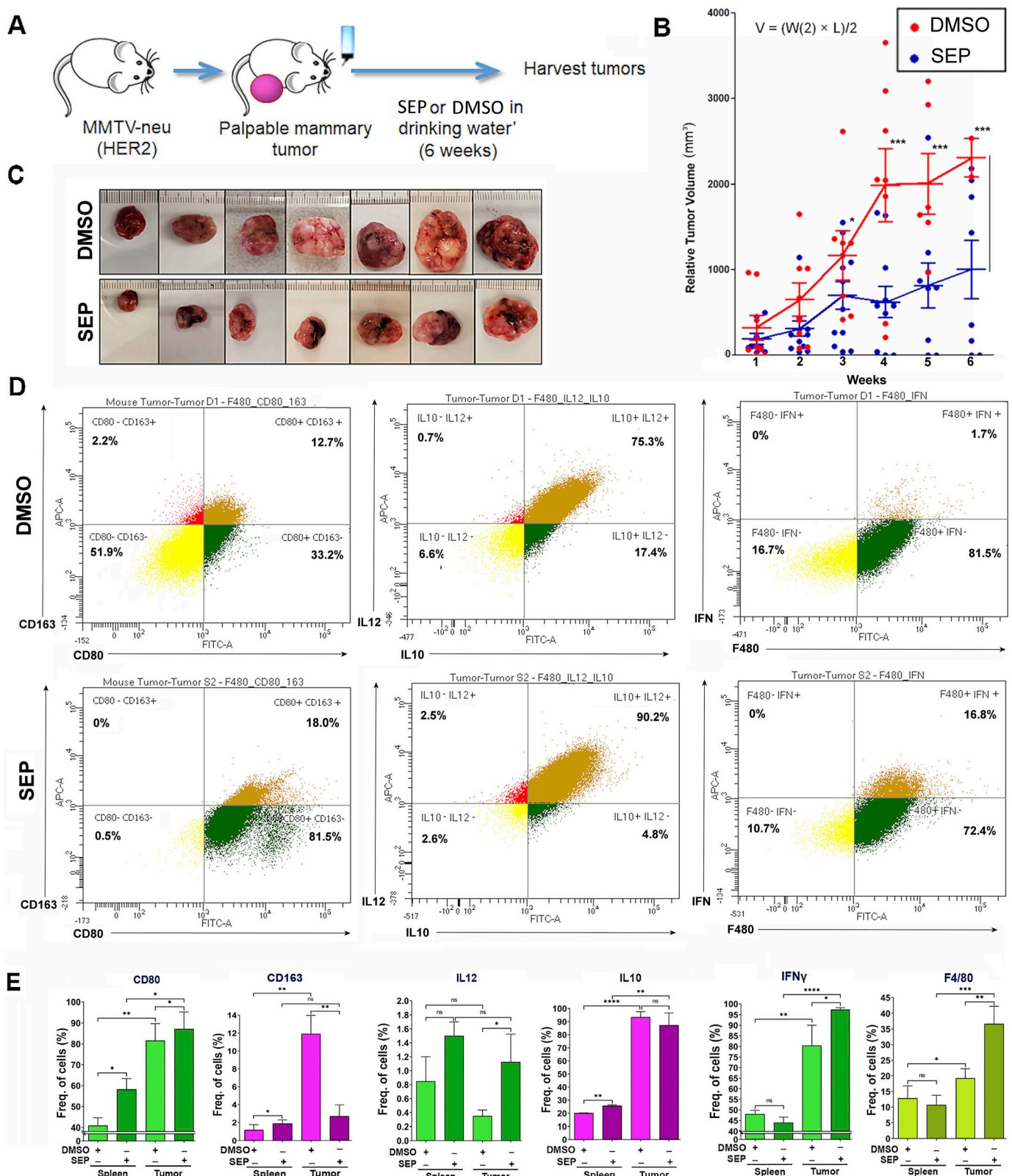

**Figure 8. Oral SEP treatment promotes the immunogenicity of TAMs and suppresses the growth of spontaneous MMTV-neu (HER2) mammary tumors.**
**(A)** Scheme of the experiment where MMTV-neu (unactivated) mice were allowed to develop palpable mammary tumors (tumor latency of 6–14 mo) and given DMSO or SEP (10 mg/kg) in drinking water ad libitum for 6 wk (n = 7). **(B)** Tumor growth was measured by caliper, and the volume was determined (V = (W(2) x L)/2). **(C)** Pictures of exercised tumors. **(D)** Exercised tumors were analyzed for M1- versus M2-TAM profiles (CD80 versus CD163; IL12 versus IL10; and IFNγ) by FACS. (Top row) DMSO-treated

dementia, menopause, aging, and inherited disorders ([74](ref)). Our study findings also strongly suggest that clinical uses of $BH_4$ and SEP for cancer treatment would warrant further investigation.

# Materials and Methods

### Cell lines

A human monocytic cell line THP-1 (Cat. No. TIB-202), and SKBR3 (Cat. No. HTB-30) and BT-474 (Cat. No. HTB-20) cancer cell lines were purchased from the American Type Culture Collection. CA1d breast cancer cells were obtained from Karmanos Cancer Institute (Detroit, MI) under Material Transfer Agreement. PBMCs were obtained from STEMCELL Technologies and AllCells, LLC. Mouse HER2-positive NT2.5 cells derived from mammary tumors of MMTV-neu/FVB mice were kindly provided by Dr. Elizabeth M Jaffee at Johns Hopkins University under Material Transfer Agreement ([57](ref)).

### Cell culture and reagents

THP-1 cells were maintained at a density of $1 \times 10^6$ cells/ml in RPMI 1640 medium (Cat. No. 11835055; Thermo Fisher Scientific) supplemented with 10% FBS, 1% penicillin–streptomycin, 2 mM GlutaMAX, 10 mM Hepes buffer, 45 g/liter glucose, and 1 mM sodium pyruvate (Cat. No. 15140122, Cat. No. 35050061, Cat. No. SH3023701, Cat. No. A2494001, and Cat. No. 11-360-070; Thermo Fisher Scientific). SKBR3 and BT-474 cells were cultured in McCoy's 5A medium with 10% FBS and 1% penicillin–streptomycin, and CA1d cells were cultured in DMEM/F12 medium with 5% horse serum, 1% penicillin–streptomycin, hydrocortisone, cholera toxin, and insulin (Cat. No. H-0888, Cat. No. C8052-2MG, and Cat. No. I1882; Sigma-Aldrich, Inc.). NT2.5 cells were cultured in RPMI-1640 medium supplemented with 20% FBS, 1.2% Hepes, 1% GlutaMAX, 1% non-essential amino acids, 1% sodium pyruvate, 1% penicillin–streptomycin, and bovine insulin (10 mg/ml) ([57](ref)). All the cells were maintained in a 37°C humidified incubator with 5% $CO_2$.

### Modulation of arginine metabolism

For induction of NO, we used NO donors: S-nitroso-N-acetylpenicillamine (SNAP, 10 or 20 $\mu$M, Cat. No. N7892; Thermo Fisher Scientific); and S-nitrosoglutathione (GSNO, 100 or 200 $\mu$M, Cat. No. sc-200349B; Santa Cruz Bio), or SEP, a precursor of NOS cofactor tetrahydrobiopterin (20 or 100 $\mu$M; Career Henan Chemical Co). For NO inhibition, we used N$\omega$-nitro-L-arginine methyl ester hydrochloride (L-NAME, 2.5 mM, Cat. No. N5751-10G; Sigma-Aldrich); the NOS2 inhibitor: 1400W hydrochloride (100 $\mu$M, Cat. No. 81520; Cayman Chemical); or the NO scavenger: 4-carboxyphenyl-4,4,5,5-tetramethylimidazoline-1-oxyl-3-oxide (cPTIO, 50 or 100 $\mu$M, Cat. No. ALX-430-001-M050; Enzo). For induction of PA, we treated cells with spermine (5 or 10 mM, Cat. No. 55513-100MG; Sigma-Aldrich). For

inhibition of PA synthesis, we used the polyamine analog, N1,N11-diethylnorspermine tetrahydrochloride (DENSPM, 50 or 100 $\mu$M, Cat. No. 0468/10; Tocris). For inhibition of arginase 1 (Arg1), we used N-hydroxy-nor-L-arginine (nor-NOHA, 50 $\mu$M, Cat. No. 10006861; Cayman Chemical); for inhibition of ornithine decarboxylase 1, we used difluoromethylornithine (DFMO, 50 $\mu$M, Cat. No. D193-25MG; Sigma-Aldrich).

### In vitro model of TAMs

#### THP-1–derived model
Human monocyte THP-1 cells were seeded at a density of $3 \times 10^5$ cells/ml and treated with 100 ng/ml PMA (Cat. No. tlrl-pma; Invi-voGen) for 24 h for their differentiation into nascent (M0) macrophages. M0 cells were then serum-starved for 2 h in an X-VIVO hematopoietic cell medium (Cat. No. BEBP04-744Q; Lonza). For M1 polarization (M1-TAMs), M0 cells were treated with PMA (100 ng/ml), 5 ng/ml LPS (Cat. No. L4391-1MG; Sigma-Aldrich), and 20 ng/ml IFNγ (Cat. No. 300-02; PeproTech) for 66 h. For M2 polarization (M2-TAMs), M0 cells were treated with PMA (100 ng/ml), 20 ng/ml IL4 (Cat. No. 200-04; PeproTech), and 20 ng/ml IL13 (Cat. No. 200-13; PeproTech) for 66 h.

### PBMC-derived model

PBMCs were plated at a density of $1.5 \times 10^6$ cells/ml in RPMI serum-free media and incubated for 3 h. Monocytes were isolated based on their adhesion to plastic. Serum-free media were aspirated to remove non-adherent cells. The adherent cells were replenished with RPMI media containing 20% FBS and incubated for 24 h. Non-adherent cells were further removed by washing with prewarmed RPMI medium. To induce M0 differentiation, the remaining adherent cells were treated with 50 ng/ml recombinant SCF, and 50 ng/ml GM-CSF (for M1 polarization) or 50 ng/ml M-CSF (for M2 polarization) (Cat. No. 573904, Cat. No. 572903, and Cat. No. 574804; BioLegend) for 7 d with 50% medium replenishment every 3 d. Upon differentiation, M0 cells were starved for 2 h as described above and given M1 or M2 polarization treatment as described above for 3 d.

### Reprogramming of M2-TAMs to M1-TAMs

THP-1 and PBMC-derived M2-TAMs were treated with 100 $\mu$M SEP (CAS No. 17094-01-8; Career Henan Chemical Co.) every day for 3 d for reprogramming to M1-TAMs. M2-TAMs were also treated with DMSO (vehicle) and 5 ng/ml LPS and 20 ng/ml IFNγ as negative and positive controls, respectively.

### PBMC-derived T-cell culture

PBMCs were incubated with 100 $\mu$g/ml DNase I solution (Cat. No. 7900; STEMCELL Technologies) at RT for 15 min and filtered through a 37-$\mu$m cell strainer. The filtered single-cell suspension was

---

tumors; (bottom row) SEP-treated tumors. **(D, E)** Quantification of the expression of M1- versus M2-TAM markers as in (D) in exercised tumors (n = 6) in comparison with spleens (n = 6) of the same animals. Error bars: ± SEM. *$P \leq 0.05$; **$P \leq 0.01$; ***$P \leq 0.001$; and ****$P \leq 0.0001$.

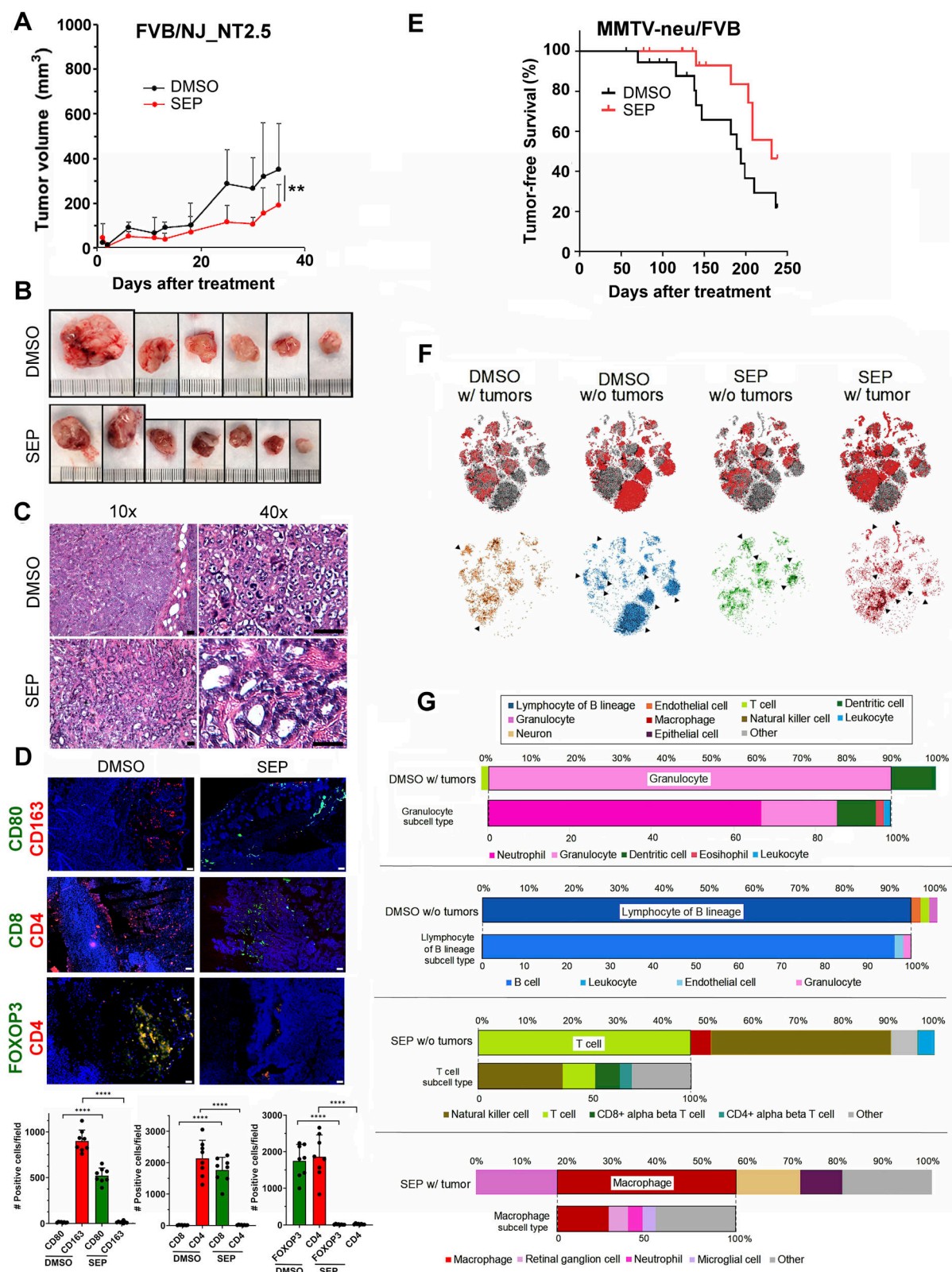

**Figure 9. Suppressive effects of oral SEP treatment on MMTV-neu (HER2) tumors are linked to the elevation of anti-tumor lymphocytes in the circulation and within tumors.**
**(A)** Mammary tumor growth of NT2.5 cells (derived from MMTV-neu tumors (57)) orthotopically transplanted into FVB/NJ mice (n = 6) and treated with DMSO or SEP (1 mg/kg) in drinking water ad libitum. Tumor growth was measured by caliper, and the volume was determined (V = (W(2) x L)/2). Statistical significance was evaluated with a two-

centrifuged at 800$g$ for 5 min, and the collected pellet was resuspended at a density of 5 × 10$^7$ cells/ml in the suspension medium composed of 2% FBS and 1 mM EDTA in PBS. T cells were then isolated using EasySep Human T Cell Isolation Kit and EasySep Magnet (Cat. No. 17951 and Cat. No. 18000; STEMCELL Technologies) following the manufacturer's protocol. Isolated T cells were seeded at a density of 1 × 10$^6$ cells/ml in RPMI medium supplemented with 10 ng/ml human recombinant IL2 (Cat. No. 78145.2; STEMCELL Technologies), 1 µg/ml anti-human CD3 antibody (Clone OKT3, Cat. No. 317326; BioLegend), and 1 µg/ml anti-human CD28 antibody (Clone CD28.2, Cat. No. 302934; BioLegend). The cells were maintained in a 37°C, 5% $CO_2$ humidified incubator for up to three subcultures with medium change every 3 d.

### SEM

THP-1–derived TAM subsets were plated at a density of 1 × 10$^6$ cells/ml on cellQART 12-well cell culture inserts with 0.4 µm PET membrane (Cat. No. 9310412; STERLITECH) and treated with SEP and relevant controls. Upon treatment, the membranes were washed with 1X PBS and fixed with 4% PFA. Then, the membranes were washed with deionized (DI) water and dehydrated using increasing concentrations of ethanol ranging from 25% to 100%. Afterward, the membranes were washed with 50% hexamethyldisilazane: ethanol solution followed by 100% hexamethyldisilazane. Later, the membranes were removed from the inserts, bound to the SEM sample stub with a carbon tape, and sputter-coated with gold for 15 s. The samples were then imaged using JEOL 7500F Scanning Electron Microscope.

### Phalloidin staining

THP-1 cells were seeded at a density of 1 × 10$^6$ cells/ml onto 12-mm coverslips (Cat. No. 1217N67; Thomas Scientific), precoated with poly-D-lysine (Cat. No. A003E; Thermo Fisher Scientific) and PureCol (Cat. No. 5005-100ML; Advanced BioMatrix), and singly placed into each well of 12-well plates. These cells were given SEP of control treatment and washed in 1X PBS and fixed with 3.7% formaldehyde in PBS for 10 min at RT. Cells were permeabilized with 0.1% Triton X-100 for 5 min and washed twice with PBS. After blocking in PBS containing 1% BSA for 30 min, cells were incubated for 20 min at RT with the phalloidin staining solution: Alexa Fluor 488 phalloidin (Cat. No. A12379; Thermo Fisher Scientific) and 1% BSA in PBS. Cells were washed with PBS and counterstained with the 4',6-diamidino-2-phenylindole (DAPI) nuclear stain for 15 min at RT. Coverslips were then mounted on slides using Fluoromount-G Mounting Medium (Cat. No. 50-187-88; Thermo Fisher Scientific) and sealed with nail polish. Fluorescence images were captured using a Zeiss Axio Observer 7 microscope.

### Immunofluorescence staining

THP-1 cells were plated on glass coverslips, polarized to TAMs, and given drug treatments as described above. Upon treatment, the coverslips were washed with 1X PBS, and cells were fixed in 4% PFA for 20 min at RT. The cells were permeabilized with 0.1% Triton X-100 for 15 min at RT and blocked with IF buffer (1.3 M NaCl, 70 mM Na2HPO4, 35 mM NaH2PO4, 77 mM NaN3, 1% BSA, 2% Triton X-100, and 5% Tween-20) containing 10% goat serum for 1 h. Cells were washed with PBS and incubated overnight at 4°C with primary antibody solution: antibody in IF buffer containing 3% saponin (Cat. No. 55-825-5100GM; Thermo Fisher Scientific). Antibodies used were as follows: TNF$\alpha$ (Cat. No. MA523720; Thermo Fisher Scientific), NOS2 (Cat. No. PA1-036; Thermo Fisher Scientific), CD163 (Cat. No. ab182422; Abcam), CD206 (Cat. No. NPB1-90020; Novus Biologicals), IL10 (Cat. No. MAB9184-100; R&D Systems), and Dectin-1 (Cat. No. PA583996; Thermo Fisher Scientific). Cells were then washed with PBS twice and incubated with secondary antibody solution (1:200 dil) for 2 h followed by DAPI stain for 15 min at RT. Coverslips were then mounted onto slides as described above. Fluorescence images were captured using an Olympus DP80 microscope and analyzed using ImageJ software.

### Immunoblotting

Cell lysates were prepared using the following lysis buffer: 25 mM Tris–HCl (pH 8), 150 mM NaCl, 1 mM EDTA (pH 8), 1% NP-40, 5% glycerol, 1X PhosSTOP (Cat. No. 4906837001; Sigma-Aldrich), and 1X Protease Inhibitor Cocktail (Cat. No. 78425; Thermo Fisher Scientific). A total protein concentration of the cell lysates was quantified using Pierce BCA Protein Assay Kit (Cat. No. 23227; Thermo Fisher Scientific). Cell lysates were mixed with the sample buffer (with beta-mercaptanol) and boiled at 100°C for 10 min. Proteins were separated by sodium dodecyl sulfate–polyacrylamide gel electrophoresis (SDS–PAGE). The separated proteins were then electroblotted to methanol-activated polyvinylidene fluoride (PVDF) membranes (Cat. No. IPVH00010; Sigma-Aldrich). Upon transfer, membranes were blocked with 5% non-fat milk in Tris-buffered saline containing 0.1% Tween-20 (TBST) and incubated overnight with the following primary antibodies: TLR2 (Cat. No. MA532787; Thermo Fisher Scientific), STAT1 (Cat. No. 9172T; Cell Signaling Technology), pSTAT1S727 (Cat. No. 8826S; Cell Signaling Technology), CD163 (Cat. No. ab182422; Abcam), CD206 (Cat. No. NPB1-90020; Novus Biologicals), STAT3 (Cat. No. 9139S; Cell Signaling

---

way ANOVA test. Error bars: ± SEM. **$P \leq 0.01$. **(B)** Pictures of exercised tumors. **(C)** Eosin-and-hematoxylin–stained tumor sections. Note the presence of numerous mammary gland–like structures in SEP-treated tumors, but not in DMSO-treated tumors. **(D)** (Top) Tumor sections were stained for M1- versus M2-TAMs (CD80 versus CD163); CD8$^+$ versus CD4$^+$ T cells, and Treg cells (CD4+/FOXOP3) by immunohistochemistry. (Bottom) The numbers of positive cells for each marker were counted per field (10x objective, n = 8–10). Error bars: ± SEM. *$P \leq 0.05$; **$P \leq 0.01$; ***$P \leq 0.001$; and ****$P \leq 0.0001$. **(E)** Percentages of tumor-free mice after DMSO versus SEP treatment. Four-week-old female MMTV-neu/FVB (n = 20) mice were treated with DMSO versus SEP (1 mg/kg) in acidified drinking water for over 7 mo. **(F)** At the end of the experiment, PBMCs were isolated from four groups (DMSO with tumors, DMSO without tumors, SEP with tumors, and SEP without tumors) and analyzed by single-cell sequencing—tSNE images comparing cell clusters among the four groups. Differentially represented clusters for each group are marked with arrows. **(G)** For each group, the major cell type and the subcell types of the former are shown. Note the predominance of T cells in the SEP without tumor group in contrast to the predominance of granulocytes in the DMSO with tumor group.

Technology), pSTAT3Y705 (Cat. No. 9145S; Cell Signaling Technology), β-actin (Cat. No. A1978; Sigma-Aldrich), and GAPDH (Cat. No. 2118S; Cell Signaling Technology). Then, they were incubated with HRP-conjugated sheep anti-mouse IgG (Cat. No. NA931-1ML; GE Healthcare Life Sciences), donkey anti-rabbit IgG (Cat. No. NA934-1ML; GE Healthcare Life Sciences), or donkey anti-goat IgG (Cat. No. A16005; Thermo Fisher Scientific) secondary antibodies (1:5,000 dil). Next, the blots were developed with SuperSignal West Dura Extended Duration Substrate (Cat. No. 34076; Thermo Fisher Scientific) and imaged using the Syngene G:BOX F3 gel doc system.

## Flow cytometry (FACS) analysis of cell surface markers

Cells were dissociated from the plates through incubation with PBS containing 5 mM EDTA for 15 min at 37°C, followed by gentle scraping. Cells were collected into 96-well V-bottom plates (Cat. No. 5665-1101; USA Scientific) and centrifuged at 1,000 rpm for 5 min. Cell pellets were washed with FACS buffer (PBS, 2% FBS) and blocked for 30 min on ice in the blocking buffer: 2% FBS, 2% goat serum, 2% rabbit serum, and 10 μg/ml human immunoglobulin G (IgG). Cells were then incubated with fluorochrome-labeled antibodies prepared in FACS buffer for 1 h on ice. The antibodies used are as follows: CD68 (Cat. No. 333821; BioLegend), CD40 (Cat. No. 334305; BioLegend), CD80 (Cat. No. 305205; BioLegend), CD163 (Cat. No. 333609; BioLegend), CD206 (Cat. No. 321109; BioLegend), anti-mouse F4/80 (Cat. No. 123118; BioLegend), anti-mouse CD80 (Cat. No. 104706; BioLegend), anti-mouse CD163 (Cat. No. 155306; BioLegend), anti-mouse NK1.1 (Cat. No. 61-5941-80; Thermo Fisher Scientific), and anti-mouse NKp46 (Cat. No. FAB2225F-025; R&D Systems). Cells were washed twice with FACS buffer and resuspended in Dulbecco's phosphate-buffered saline containing 2% formaldehyde. Samples were assayed on the BD FACSCanto II system, followed by analyses with FlowJo software.

## FACS analysis of intracellular markers

Cells were treated with 1 μg/ml Brefeldin A (Cat. No. B7450; Thermo Fisher Scientific) for 5 h at 37°C. Upon treatment, cells were collected into 96-well V-bottom plates and centrifuged as described above. The samples were then incubated with 1X fixation/permeabilization buffer (Cat. No. FC007; R&D Systems) for 12 min at 4°C. After fixation, samples were centrifuged at 1,600 rpm for 5 min and washed with the following permeabilization/washing buffer: PBS, 2% FBS, and 0.1% Triton X-100. Samples were then blocked with blocking buffer containing 0.1% Triton X-100 for 10 min. After blocking, samples were incubated with fluorochrome-labeled antibodies prepared in permeabilization/washing buffer for 45 min at 4°C. The following antibodies were used: anti-mouse IFNγ (Cat. No. IC485F-025; R&D Systems), anti-mouse IL12 (Cat. No. 505206; BioLegend), anti-mouse IL10 (Cat. No. 505006; BioLegend), and anti-mouse perforin (Cat. No. 11-9392-80; Thermo Fisher Scientific). After antibody incubation, samples were washed with permeabilization/washing buffer, resuspended in Dulbecco's phosphate-buffered saline containing 2% formaldehyde, and assayed on the BD FACSCanto II system, followed by analyses with FlowJo software.

## Measurement of BH₄ production

THP-1–derived TAMs plated in six wells were used for the measurement of BH4. Cells were washed with ice-cold PBS to remove remaining media. Upon washing, cells were scraped gently, and the cell pellets were collected into Eppendorf tubes. The cell pellets were vortexed for 10 s and flash-frozen in liquid nitrogen (N2). Then, the pellets were thawed at RT. The process was repeated five times. Then, the pellets were centrifuged at 1,000 rpm for 5 min, and the supernatants were collected into fresh tubes. The BH4 level in cell lysate was measured using an ELISA kit (Cat. No. abx354211; Abbexa) following the manufacturer's protocol. Cellular BH4 levels were normalized using the total protein concentration in cell lysates.

## Live-cell imaging of NO and PA

Intracellular NO and PA levels were detected in live cells with the help of cell-permeable fluorescent dyes. Cells grown in phenol red and serum-free media were stained with 20 μM diaminorhodamine-4M acetoxymethyl ester (DAR-4M AM) (Cat. No. ALX-620-069-M001; Enzo) or 20 μM PolyamineRED (Cat. No. FDV-0020; DiagnoCine) according to the manufacturers' protocol to detect NO and PA, respectively. Fluorescence images were captured using an Olympus DP80 microscope and analyzed using ImageJ software.

## Measurement of NO levels

Nitric Oxide Fluorometric Assay Kit (Cat. No. K252; BioVision) was used to quantify the NO levels present in the CM. The samples were filtered through a 10-kD cutoff Microcon filter (Millipore) to remove proteins present in the CM. The flow-through was reacted with the assay reagents in the dark according to the manufacturer's protocol, and the fluorescence intensity was measured at the wavelengths of Ex/Em = 360/450 nm.

## Measurement of PA levels

PA levels in the CM were quantified using Fluorometric Total Polyamine Assay Kit (Cat. No. K475; BioVision) according to the manufacturer's protocol with modifications. This kit determines the level of hydrogen peroxide produced through oxidation of polyamines by spermine/spermidine oxidase in the kit. To remove high background levels of hydrogen peroxide produced by macrophages before the assay, the CM were pretreated with catalase (Cat. No. C40-100MG; Sigma-Aldrich) at 100 μg/ml and incubated at 37°C for 1.5 h. Proteins were precipitated with Sample Clean-up Solution provided by the kit and removed by filtration through a 10-kD cutoff Microcon filter. The flow-through was reacted with the assay reagents in the dark according to the manufacturer's protocol, and the fluorescence intensity was measured at the wavelengths of Ex/Em = 535/587 nm.

## Measurement of cytokine secretion

Cells were cultured in RPMI serum-free media for 2 d after treatment, and the CM were collected. Secreted cytokines in the CM were

quantified using ELISA kits according to the manufacturers' protocol. The ELISA kits used were as follows: IL12 (Cat. No. D1200; R&D Systems), IL6 (Cat. No. D6050; R&D System), IL1$\beta$ (Cat. No. DLB50; R&D System), TNF$\alpha$ (Cat. No. DTA00D; R&D System), IL10 (Cat. No. D1000B; R&D Systems), and TGF$\beta$ (Cat. No. ab108912; Abcam).

## Antigen presentation assay

THP-1–derived TAMs were detached from the plates, collected into 96-well V-bottom plates, and centrifuged as described before. The cells were then pulsed with 20 $\mu$g/ml OVA323–339 peptide (Cat. No. RP10610-1; GenScript) for 2 h at 37°C and 5% CO2. Cells were then washed twice with FACS buffer (see above) to remove residual OVA peptides and incubated with blocking buffer for 30 min at 4°C. Samples were stained with fluorescein isothiocyanate (FITC)–labeled HLA-DR antibody (Cat. No. 307604; BioLegend) in FACS buffer for 1 h at 4°C and analyzed by flow cytometry.

## TAM, T-cell, and cancer cell co-culture model

A triple co-culture model of breast cancer cells, TAMs, and T cells was developed to emulate and study the tumor–immune interactions within the TME.

## Transwell co-culture system

THP-1 cells were seeded on cellQART 12-well inserts (0.4 $\mu$m pore size and PET membrane) at a density of 3 × 10$^5$ cells/ml and subjected to differentiation, polarization, and reprogramming as described above. After establishment of macrophages, the inserts were washed with fresh RPMI media, and breast cancer cells (CA1d, SKBR3, or BT-474) in RPMI media were seeded on top of THP-1 cells at a ratio of 1:2 (cancer cells: THP-1 cells). Into separate bottom wells of 12-well plates, the equal amounts of breast cancer cells were seeded in RPMI media. Once cells were attached, PBMC-derived T cells were seeded on top of cancer cells at a ratio of 1:2 (cancer cell: T cell). The inserts containing cancer cells and THP-1 were placed on top of the wells containing cancer cells and T cells. The co-cultures were incubated in a humidified incubator set at 37°C with 5% CO2 for further experiments.

## Direct co-culture system

Monocytes and T cells isolated from the same PBMCs were used to establish the direct co-culture system. Monocytes were seeded on 12-well plates at a density of 1.5 × 10$^6$ cells/ml, differentiated, polarized into relevant TAM subsets, and reprogrammed as described above. The media were removed, and cells were washed with fresh RPMI serum-free media. Breast cancer cells and T cells were sequentially added to the 12-well plates at a ratio of 1:2:2 (cancer cell: TAM: T cell) in the same RPMI media. The co-culture was incubated in a humidified incubator set at 37°C with 5% CO$_2$ for further experiments.

## T-cell proliferation assay

PBMC-derived T cells were washed with PBS and resuspended at a density of 1 × 10$^6$ cells/ml. The cells were then stained with the CellTrace CFSE dye (Cat. No. C34554; Thermo Fisher Scientific) following the manufacturer's protocol. CFSE-labeled T cells were directly co-cultured with PBMC-derived TAMs and cancer cells as described above and incubated for 4 d. T cells were isolated and stained with fluorochrome-labeled anti-human CD3 (Cat. No. 317342; BioLegend) and anti-human CD8a (Cat. No. 301014; BioLegend) antibodies as described above. The cytotoxic T-cell proliferation, indicated by changes in CFSE signals, was measured by flow cytometry.

## Measurement of IFN$\gamma$ production

The triple co-culture, composed of cancer cells, TAMs, and T cells, was established as described above and incubated for 2 d. After incubation, the cells were treated with 1 $\mu$g/ml Brefeldin A for 4 h to block protein secretion. T cells were isolated and stained with anti-human CD3 and CD8a antibodies (surface staining) followed by intracellular staining with anti-human IFN$\gamma$ (Cat. No. IC285F-100; R&D Systems) antibody, as described before, and analyzed by flow cytometry.

## Degranulation assay

CD107a marker expression was measured to analyze the degranulation of cytotoxic T cells. The triple co-culture system was set up as previously described and incubated for 2 d. Upon incubation, 10 $\mu$l of anti-human CD107a antibody (Cat. No. 328606; BioLegend) was added to each well and incubated in a humidified incubator set at 37°C with 5% CO$_2$. After 1 h of incubation, 1X eBioscience Monensin Solution (Cat. No. 00-4505-51; Thermo Fisher Scientific) was added to each well and the incubation was continued for three more hours to block protein transport. T cells were then isolated and stained with CD3 and CD8a antibodies, as previously described. CD107a expression in cytotoxic T cells was measured by flow cytometry.

## Measurement of extracellular ATP

During ICD—a form of apoptosis—dying cells release ATP to induce immunogenic phagocytosis. Thus, the increase in secreted ATP is used to determine the occurrence of ICD. A luminescent ATP detection assay kit (Cat. No. ab113849; Abcam) was used to quantify ATP released from cells. The co-culture system was set up as described above, and 100 $\mu$l of CM from the bottom well (seeded with cancer cells and T cells) was analyzed for ATP levels, in comparison with CM of monocultured TAMs, T cells, and cancer cells as controls. ATP concentration was quantified using the standard following the manufacturer's protocol.

## Cell cycle analysis

Cancer cells were labeled with CellTracker Green CMFDA Dye (Cat. No. C7025; Thermo Fisher Scientific) according to the manufacturer's

protocol. The labeled cancer cells were directly co-cultured with PBMC-derived TAMs and T cells as described above and incubated for 4 d. Cells were harvested using 0.05% Trypsin (Cat. No. 25-300-062; Thermo Fisher Scientific), washed with PBS, and centrifuged at 1,000 rpm for 5 min. The pelleted cells were fixed with precooled 70% ethanol for 2 h at 4°C. Cells were centrifuged at 4,000 rpm for 2 min, resuspended in PBS containing 0.25% Triton X-100, and incubated on ice for 15 min. Cells were centrifuged and resuspended in PBS containing 10 $\mu$g/ml ribonuclease (RNase) A and 20 $\mu$g/ml propidium iodide (PI) (Cat. No. R4642-10MG and Cat. No. P4170-10MG; Sigma-Aldrich) for the staining of DNA. Cells were transferred to FACS tubes, incubated in the dark for 30 min at RT, and analyzed for cell cycle profiles by flow cytometry.

### Apoptosis assay

The direct co-culture model was set up using CMFDA-labeled cancer cells as described before and incubated for 4 d. Adherent cells were harvested, washed with FACS buffer, and centrifuged at 1,000 rpm for 5 min. The pelleted cells were then resuspended at a density of 1 × 10$^6$ cells/ml in Annexin V Binding Buffer: 10 mM Hepes, 150 mM NaCl, and 2.5 mM CaCl2 in PBS. Resuspended cells were transferred to FACS tubes, stained with 5 $\mu$l of Annexin V (Cat. No. A35110; Thermo Fisher Scientific) and 20 $\mu$g/ml of PI, and incubated in the dark for 15 min at RT. Each tube was replenished with 400 $\mu$l of Annexin V Binding Buffer and analyzed by flow cytometry.

### Animal study

All in vivo experiments were performed in compliance with the Guide for the Care and Use of Laboratory Animals (National Research Council, National Academy Press, Washington, D.C., 2010) and with the approval of the Institutional Animal Care and Use Committee of the University of Toledo, Toledo, OH (Protocol No: 108658), and Case Western Reserve University, Cleveland, OH (Protocol No. 2022-0080).

For the tumor treatment study, 2-mo-old female MMTV-neu/FVB (unactivated) (n = 14) mice were obtained from the Jackson Laboratory (ID. IMSR_JAX:002376), housed under regular conditions, and given ad libitum access to acidified water and regular chow. The mice were maintained until spontaneous mammary tumors became palpable (~5 mm long, 6–14 mo). Upon tumor onset, mice were divided into vehicle (DMSO) versus SEP (10 mg/kg) treatment groups (70, 71). The drugs were dissolved in acidified drinking water and administered to mice ad libitum for 6 wk (75). The tumor growth, body weight, and the morbidity of the animals were monitored twice a week. Tumor volume was calculated by the modified ellipsoidal formula: Volume = (Length x Width x Width) x ½. At the end of treatment, mice were euthanized, and mammary tumors were processed for further analyses as described below. As a complementary model, 6-wk-old FVB/NJ mice (n = 20) were obtained from the Jackson Laboratory and treated with DMSO or SEP (10 mg/kg) through drinking water until they became 5 mo old. These animals were injected with NT2.5 cells derived from mammary tumors of MMTV-neu/FVB (57) at around 2.5 × 10$^6$ cells/mouse through nipples and treated with DMSO or SEP for another 100 d. The tumor growth, body weight, and the morbidity of the animals were monitored

three times a week. At the end of treatment, mice were euthanized, and mammary tumors were processed for further analyses.

For the tumor prevention study, 4-wk-old female MMTV-neu/FVB (unactivated) (n = 20) mice were divided into the vehicle (DMSO) versus SEP (1 mg/kg) treatment group. The drugs were dissolved in acidified drinking water and administered to mice ad libitum for 7 mo. Tumor incidence was monitored, and the percentage of tumor-free mice was quantified. The body weight, tumor incidence, latency, and size were monitored twice a week, and urine and fecal samples were collected once a week. At the end of treatment, mammary tumors, spleens, livers, and blood were harvested and processed for further analyses.

### Profiling of macrophage cells in tumors and spleens

Freshly harvested tumors and spleens were processed for the profiling of resident macrophages. Tumors and spleens were weighed and reacted in the digestion mixture (10 ml/g of tissue, 3 mg/ml collagenase A [Cat. No. 10103578001; Sigma-Aldrich], and 25 $\mu$g/ml DNase I [Cat. No. 10104159001; Sigma-Aldrich]) in HBSS (Cat. No. 14025092; Thermo Fisher Scientific) with gentle motions on a platform shaker for 45 min at 37°C (76). To stop the enzymatic digestion, the samples were treated with 10 ml of staining buffer (1% [wt/vol] BSA in PBS). The cell suspension was then filtered through a 100-$\mu$m cell strainer, and the volume was adjusted to 20 ml with staining buffer. Then, cells were pelleted by centrifugation at 500$g$ for 7 min at 4°C. To remove red blood cells, the pelleted cells were suspended in 3 ml of 1X RBC lysis buffer (Cat. No. 00-4300-54; Thermo Fisher Scientific) and incubated on ice for 10 min. A volume of 30 ml of staining buffer was added, and cells were centrifuged and resuspended in 1 ml of staining buffer. To block Fc receptors (to avoid unwanted antibody binding), cells were treated with 2 $\mu$l of anti-mouse CD16 (FcγII)/CD32 (FcγIII) antibody (Cat. No. 14-0161-82; Thermo Fisher Scientific) and incubated on ice for 30 min with mixing at 10-min intervals. A volume of 4 ml of staining buffer was added, and samples were centrifuged at 500$g$ for 5 min at 4°C. Then, cell pellets were resuspended in 1 ml of staining buffer (77). Cells were then stained with relevant fluorochrome-labeled antibodies, as previously described, and analyzed by flow cytometry.

### PBMC isolation from mouse blood

At the end of the tumor prevention experiment, MMTV-neu/FVB mice were grouped into four groups (DMSO with tumors, DMSO without tumors, SEP with tumors, and SEP without tumors). Blood was collected from submandibular veins, pooled by group, and mixed with the equal volume of PBS-EDTA solution (Cat. No. J60893.K3; Thermo Fisher Scientific). To isolate mononuclear cells, the diluted blood was added on top of Lymphoprep medium (Cat. No. 07851; STEMCELL Technologies) within a SepMate-15 centrifuge tube (Cat. No. 85415; STEMCELL Technologies) and spun at 1,200$g$ for 10 min at room temperature for density gradient centrifugation. Mononuclear cells accumulated at the interface between the top serum and bottom Lymphoprep layers were carefully collected and transferred into a separate centrifuge tube. Cells were washed in PBS plus 2% FBS twice, resuspended in RPMI-1640 medium, and cryopreserved until single-cell sequencing.

### Single-cell sequencing

Single-cell sequencing was performed with GEM-X, Chromium Single Cell Gene Expression (3′ GEX V3.1) chips on 10X Genomics Chromium X Processor at the Discovery Lab in the Global Center for Immunotherapy and Precision Immuno-Oncology at Lerner Research Institute, Cleveland Clinic. Fastq files were mapped to the GRCh38 reference human genome using Cellranger (v5.0.0) (78). Cells containing less than 600 genes and/or more than 30% mitochondrial and ribosomal genes were removed. Sample-specific Seurat objects were created using Seurat (v4.3.0) (79), then normalized using Seurat's SCTransform method. Samples were integrated based on variable features using Seurat's IntegrateData function. To help predict cell types, the Seurat object was uploaded to BioTuring BBrowserX (80 *Preprint*), and cells were annotated using their deep learning–based mouse cell–type prediction model (subcell type [version.2] model).

### Immunohistochemistry

To determine the expression of specific markers in tissues harvested from mice, paraffin-embedded sections were analyzed by immunohistochemistry. Briefly, sections were deparaffinized, hydrated, and treated with antigen unmasking solution (Cat. No. H-3300-250, pH 6.5; Vector Lab., Inc.) or with Tris–EDTA buffer (10 mM Tris base, 1 mM EDTA solution, and 0.05% Tween-20, pH 9.0), which had been heated to 95–100°C in a pressure cooker. After being blocked with non-immune goat serum, sections were processed for immunofluorescence staining as described below.

### Immunofluorescence staining and imaging

Immunofluorescence staining/imaging was performed as described previously (21, 81). Samples were incubated with a primary antibody for overnight at 4°C in a humidified chamber. Primary antibodies used for immunofluorescence staining of mammary tumors were antibodies against mouse CD80 (Cat. No. 104705; BioLegend), CD163 (Cat. No: ab182422; Abcam), CD8 (Cat. No. PA579011; Thermo Fisher Scientific), CD4 (Cat. No. 13-9766-82; Thermo Fisher Scientific), and FOXOP3 (Cat. No. MAB8214; Bio-Techne). After intensive washing (three times, 15 min each) in 0.1% BSA, 0.2% Triton X-100, 0.05% Tween-20, and 0.05% NaN3 in PBS, fluorescence-conjugated secondary antibodies (Molecular Probes) were added for 2 h at room temperature. Nuclei were stained with 0.5 ng/ml DAPI. After being mounted with anti-fade solution, epifluorescence imaging was performed on the Leica Thunder Imaging platform with Leica LAS X Life Science Microscope software.

### Image analysis

Quantification of fluorescence signal in micrographs captured with 10x objective was performed with ImageJ software (NIH) referring to the owner's manual (http://imagej.net/docs/guide/146.html) as described previously (21). For particle analysis to count positively stained immune cells, images were converted to gray-scale images,

and the same threshold ranges were set for each marker analysis. Using the Analyze Particles function, the numbers of positive cells were automatically counted. For each tumor section, 8–10 regions were analyzed. The mean value was represented as the number of positive cells per field. The statistical significance of the data was further evaluated using GraphPad Prism version 10 software (see the statistics section).

### Statistical data analysis

The experimental results are presented as the mean ± SEM. All the experiments were performed in replicates (n ≥ 3), and unless otherwise indicated, two-tailed $t$ tests were performed to obtain the statistical significance of the mean difference. $P$-values ≤ 0.05 were considered statistically significant. Flow cytometry data analyses were performed using FlowJo version 10.5. Imaging analyses were performed on ImageJ software. All statistical analyses were carried out using GraphPad Prism version 10.

# Supplementary Information

# Acknowledgements

We thank Raghvendra Srivastava, Vladimirand Makarov, and Ivan Juric of the Discovery Lab in the Global Center for Immunotherapy and Precision Immuno-Oncology at Cleveland Clinics Foundation for single-cell sequencing and analysis. We would also like to thank Drs. Andrea Kalinoski and David Weaver in the Imaging Core at the University of Toledo for various supports in imaging and FACS analyses. This work was supported by the startup funds from the University of Toledo Health Science Campus, College of Medicine and Life Sciences, Department of Cancer Biology and from MetroHealth Medical Center/Case Western Reserve University to S Furuta; Ohio Cancer Research Grant (Project #: 5017) to S Furuta; Medical Research Society (Toledo Foundation, #206298) Award to S Furuta; American Cancer Society Research Scholar Grant (RSG-18-238-01-CSM) to S Furuta; and National Cancer Institute Research Grant (R01CA248304) to S Furuta.

## Author Contributions

V Fernando: conceptualization, data curation, formal analysis, investigation, visualization, methodology, writing—original draft, and project administration.
X Zheng: resources, investigation, methodology, and project administration.
V Sharma: investigation, methodology, and project administration.
O Sweef: investigation, visualization, methodology, and project administration.
E-S Choi: methodology and project administration.
S Furuta: conceptualization, resources, data curation, formal analysis, supervision, funding acquisition, investigation, visualization, methodology, project administration, and writing—original draft, review, and editing.

## Conflict of Interest Statement

The authors declare that they have no conflict of interest.

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
