## [Reviewer comments · Life Science Alliance]

Life Science Alliance

Reprogramming of breast tumor-associated macrophages with modulation of arginine metabolism

Veani Fernando, Xunzhen Zheng, Vandana Sharma, Osama Sweef, Eun-Seok Choi, and Saori Furuta

DOI: <https://doi.org/10.26508/lsa.202302339>

Corresponding author(s): Saori Furuta, Case Western Reserve University and Saori Furuta, Case Western Reserve University

Review Timeline:

Submission Date:	2023-08-25
Editorial Decision:	2023-10-23
Revision Received:	2024-08-07
Editorial Decision:	2024-08-15
Revision Received:	2024-08-16
Accepted:	2024-08-19

Transaction Report:

October 23, 2023

Re: Life Science Alliance manuscript #LSA-2023-02339-T

Prof. Saori Furuta
Case Western Reserve University
Medicine
2500 Metrohealth Dr.
Cleveland, OH 44109

Dear Dr. Furuta,

Thank you for submitting your manuscript entitled "Reprogramming of breast tumor-associated macrophages with modulation of arginine metabolism" to Life Science Alliance. The manuscript was assessed by expert reviewers, whose comments are appended to this letter. We invite you to submit a revised manuscript addressing the Reviewer comments.

Thank you for this interesting contribution to Life Science Alliance. We are looking forward to receiving your revised manuscript.

Sincerely,

B. MANUSCRIPT ORGANIZATION AND FORMATTING:

Reviewer #1 (Comments to the Authors (Required)):

In this study, Fernando et al. provide a proof of concept for the therapeutic benefit of reprogramming tumor associated macrophages via the modulation of arginine metabolism.

Previously the lab has shown that M1 and M2 macrophage differentially metabolize arginine: M1 macrophages convert arginine in nitric oxide (NO) while M2 macrophages convert arginine in polyamines (PA). They hypothesize that oxidative degradation in the TME leads to degradation of BH4 co-factor and NO synthetase dysfunction and M2 polarization. Therefore authors propose that providing SEP, a precursor of BH4, could redirect arginine metabolism of macrophages towards NO synthesis.

First, authors dissect the metabolic pathway of M1 and M2 human macrophages (CD14+ monocyte derived or THP1 cell line derived). Then they explore the impact of SEP on TAM polarisation and immunosuppressive function. Lastly, they test the therapeutic potential of SEP in vivo by using a spontaneous HER2+ mammary tumor model.

Overall, the study is interesting and well conducted. The authors provide a thorough analysis of the M1 and M2 role in T cell priming. The experiments are well controlled. Lastly, authors provide a very nice proof of concept experiment by using a spontaneous tumor model for HER2+ breast cancer.

One comment: since authors have shown that M1/M2 TAMs have distinct T-cell priming abilities in vitro, authors should also provide the phenotype of T cells in the HER2 model in vivo and compare SEP treated and untreated mice. Simple flow cytometry profiling could include: CD44, CD62L, PD1, LAG3, CD103, CD69, FoxP3 for instance.

This would help understand whether SEP treatment impacts innate mechanisms or adaptive immunity.

Reviewer #2 (Comments to the Authors (Required)):

In this manuscript, Veani Fernando and collaborators explored the mechanisms by which modulation of arginine metabolism could reprogram M2-like macrophages and determined the therapeutic potentials of SEP for HER2- positive breast cancer.

In general, this is an interesting study. However, as explained in Fig 1, the protocols used in this study involved the classical way to polarized macrophages into M1-like or M2-like cells. These cells were wrongly named TAMs. To study TAMs, macrophages should be cultured with tumor-conditioned medium or extracted from tumor-bearing mice. Thus, the authors should incubate THP1 or PBMC-derived macrophages with Breast cancer-derived conditioned medium to obtain TAM-like cells. In addition, most of the experiments (including antigen presentation, and cytotoxic T cell activation) should be performed in macrophages purified from breast tumor-bearing mice model (as used in Fig 8). At this stage, all the experiments in vitro (presented in Fig 1-Fig7 of this manuscript) demonstrated the impact of SEP on M1/ M2-like cells, not TAMs.

Although the authors investigated the effect of SEP on M2-like cells and M1-like cells, to be scientific relevant, in my opinion, most of the experiments should be performed in TAMs purified from the mouse model.

Responses to Comments of the Editor and Reviewers:

We appreciate the constructive comments of reviewers. In the revised manuscript, we have performed additional experiments and revised the manuscript to address the reviewers' concerns. We have included our point-by-point responses (in blue) to the reviewers' comments (boxed) below.

Reviewer #1:

General Comments: Overall, the study is interesting and well conducted. The authors provide a thorough analysis of the M1 and M2 role in T cell priming. The experiments are well controlled. Lastly, authors provide a very nice proof of concept experiment by using a spontaneous tumor model for HER2+ breast cancer.

We appreciate the Reviewer's comment.

Specific Comments:

Since authors have shown that M1/M2 TAMs have distinct T-cell priming abilities in vitro, authors should also provide the phenotype of T cells in the HER2 model in vivo and compare SEP treated and untreated mice. Simple flow cytometry profiling could include: CD44, CD62L, PD1, LAG3, CD103, CD69, FoxP3 for instance. This would help understand whether SEP treatment impacts innate mechanisms or adaptive immunity.

We appreciate the reviewer's thoughtful suggestion. To address this comment, we have performed two new animal studies, one using a transgenic mouse model of HER2+ breast cancer, MMTV-neu; and the other using orthotopic transplantation model of HER2+ breast cancer cells, FVB-NT2.5. In the latter model, NT2.5 cells derived from MMTV-neu mammary tumors were transplanted into mammary glands of immuno-competent FVB mice. These mice were subjected to DMSO (control) or SEP treatment for 7 weeks and tested for the effect of SEP treatment on tumor growth. In contrast, in MMTV-neu model, mice were treated with DMSO or SEP for 8 months, starting at the pre-pubertal stage, and tested for the effect of SEP on suppression of tumor incidence at the age of 6-12 months (**Fig. 9**). We saw that SEP demonstrated potent anti-tumor effects in both models (**Fig. 9A, B, E**). Then, we analyzed their immunogenic profiles.

For FVB-NT2.5 model, we harvested mammary tumors of DMSO- vs. SEP-treated animals and compared their immunogenic profiles by immunohistochemistry (unfortunately, we accidentally fixed or froze all harvested tumors, making them unsuitable for FACS analysis). In DMSO-treated tumors, we saw the prominent M2-TAMs (CD163+) with little M1-TAMs (CD80+), as well as predominant CD4+ T cells and CD4+/FOXOP3+ Treg cells with little CD80+ T cells. In contrast, in SEP-treated tumors, we saw the prominent M1-TAMs (CD80+) with little M2-TAMs (CD163+), as well as predominant CD8+ T cells with little CD4+ T cells and CD4+/FOXOP3+

Treg cells (**Fig. 9D**). This result was consistent with our *in vitro* results. For MMTV-neu model, we harvested PBMC of DMSO- vs. SEP-treated animals at the end of 8 month treatment and compared the cell compositions by single cell sequencing. This analysis included 4 animal groups: DMSO with tumors (80% of DMSO group) or without tumors (20% of DMSO group); and SEP with tumors (45% of SEP group) or without tumors (55% of SEP group). DMSO with tumor group was abundant in granulocyte populations largely composed of neutrophils, whereas SEP with tumor group was abundant in macrophages which were presumably pro-tumor type. Conversely, SEP without tumor group was highly abundant in T lymphocyte populations composed of natural killer cells, effector CD8+, and CD4+ T cells. DMSO without tumor group was abundant in B lymphocytes, further suggesting the roles of anti-tumor lymphocytes in tumor prevention (**Fig. 9F, G**). These results altogether confirmed that anti-tumor effects of SEP were strongly linked to the elevated levels of anti-tumor T lymphocytes in circulation and within tumors in association with the immunogenic shift of tumor-associated macrophages. These new results are included as Figure 9 in the revised manuscript.

Reviewer #2:

General Comments:

In general, this is an interesting study.

We appreciate the reviewer's comment.

Specific Comments:

1. As explained in Fig 1, the protocols used in this study involved the classical way to polarized macrophages into M1-like or M2-like cells. These cells were wrongly named TAMs. To study TAMs, macrophages should be cultured with tumor-conditioned medium or extracted from tumor-bearing mice. Thus, the authors should incubate THP1 or PBMC-derived macrophages with Breast cancer-derived conditioned medium to obtain TAM-like cells.

We agree with the reviewer's comment and reworded "TAMs" to "M1-like and M2-like macrophages" in results of *in vitro* experiments. We did seek to establish TAMs from THP1 or PBMC-derived monocytes by co-culturing then with breast cancer cells using a transwell system to utilize the secretome of cancer cells. However, in our hand, these monocytes only became nascent (M0) macrophages instead of being polarized to M2 macrophages. Therefore, we resorted to the well established protocols utilizing combinations of cytokines to generate M1-like or M2-like macrophages from THP1 cells [1, 2] as well as PBMC in our *in vitro* experiments. We also describe the reasons why we failed to utilize TAMs isolated from tumors in our response below.

2. In addition, most of the experiments (including antigen presentation, and cytotoxic T cell activation) should be performed in macrophages purified from breast tumor-bearing mice model (as used in Fig 8). At this stage, all the experiments in vitro (presented in Fig 1-Fig7 of this manuscript) demonstrated the impact of SEP on M1/ M2-like cells, not TAMs. Although the authors investigated the effect of SEP on M2-like cells and M1-like cells, to be scientific relevant, in my opinion, most of the experiments should be performed in TAMs purified from the mouse model.

We have reworded “TAMs” to “M1- like and M2-like macrophages” in results of in vitro experiments according to the reviewer’s comment. We have also included additional two mouse experiments to test the anti-tumor, pro-immunogenic effects of SEP on TAMs in vivo (please see our response to Reviewer 1’s comment above).

We strongly agree with the reviewer’s suggestion of using TAMs purified from mouse tumors in our in vitro experiments. However, we were not able to obtain sufficient numbers of viable TAMs from tumors because of technical difficulties of this approach. First, chemical and mechanical dissociation of TAMs from tumors greatly impacted the integrity and viability of cells. These isolated viable TAMs (3-5% of the total cells) were less adherent and showed much lower cytokine production and marker expression, compared to THP-1 cells polarized in culture. Second, there were great variations in the M1/M2 TAM ratios between different tumors, and even among M2-TAM populations, there was heterogeneity in marker expression levels. Such variations necessitated additional sorting to isolate homogenous M2 TAM populations, further reducing the number of cells obtained. Because of these difficulties in obtaining TAMs from tumors, we utilized THP1 cells and PBMC to polarize them to M1-like or M2-macrophages using well established protocols [1, 2] for our in vitro experiments. The results of these in vitro experiments were well supported by the results of in vivo experiments, attesting to the validity of the in vitro models we used.

REBUTTAL REFERENCES

1. Genin M, Clement F, Fattaccioli A, Raes M and Michiels C, M1 and M2 macrophages derived from THP-1 cells differentially modulate the response of cancer cells to etoposide. *BMC Cancer* **15**: 577, 2015.
2. Mohd Yasin ZN, Mohd Idrus FN, Hoe CH and Yvonne-Tee GB, Macrophage polarization in THP-1 cell line and primary monocytes: A systematic review. *Differentiation* **128**: 67-82, 2022.

August 15, 2024

RE: Life Science Alliance Manuscript #LSA-2023-02339-TR

Prof. Saori Furuta
Case Western Reserve University
Medicine
2500 Metrohealth Dr.
Cleveland, OH 44109

Dear Dr. Furuta,

Thank you for submitting your revised manuscript entitled "Reprogramming of breast tumor-associated macrophages with modulation of arginine metabolism". We would be happy to publish your paper in Life Science Alliance pending final revisions necessary to meet our formatting guidelines.

- please be sure that the authorship listing and order is correct
- please upload all figure files as individual ones, including the supplementary figure files; all figure legends should only appear in the main manuscript file
- please upload your main manuscript text as an editable doc file
- please add the Twitter handle of your host institute/organization as well as your own or/and one of the authors in our system
- please add your main and supplementary figure legends to the main manuscript text after the references section
- please exclude figures from the manuscript text and upload them separately
- please add callouts for Figures 2D and E; 4G; 7F and S1A-B; S2A-B to your main manuscript text

A. FINAL FILES:

B. MANUSCRIPT ORGANIZATION AND FORMATTING:

Sincerely,

August 19, 2024

RE: Life Science Alliance Manuscript #LSA-2023-02339-TRR

Prof. Saori Furuta
Case Western Reserve University
Medicine
2500 Metrohealth Dr.
Cleveland, OH 44109

Dear Dr. Furuta,

Thank you for submitting your Research Article entitled "Reprogramming of breast tumor-associated macrophages with modulation of arginine metabolism". It is a pleasure to let you know that your manuscript is now accepted for publication in Life Science Alliance. Congratulations on this interesting work.

DISTRIBUTION OF MATERIALS:

Again, congratulations on a very nice paper. I hope you found the review process to be constructive and are pleased with how the manuscript was handled editorially. We look forward to future exciting submissions from your lab.

Sincerely,
